# Early intrinsic hyperexcitability does not contribute to motoneuron degeneration in amyotrophic lateral sclerosis

**Félix Leroy\*, Boris Lamotte d'Incamps, Rebecca D Imhoff-Manuel, Daniel Zytnicki**

Laboratory of Neurophysics and Physiology, UMR 8119, Paris Descartes University, Paris, France

**Abstract** In amyotrophic lateral sclerosis (ALS) the large motoneurons that innervate the fast-contracting muscle fibers (F-type motoneurons) are vulnerable and degenerate in adulthood. In contrast, the small motoneurons that innervate the slow-contracting fibers (S-type motoneurons) are resistant and do not degenerate. Intrinsic hyperexcitability of F-type motoneurons during early postnatal development has long been hypothesized to contribute to neural degeneration in the adult. Here, we performed a critical test of this hypothesis by recording from identified F- and S-type motoneurons in the superoxide dismutase-1 mutant G93A (mSOD1), a mouse model of ALS at a neonatal age when early pathophysiological changes are observed. Contrary to the standard hypothesis, excitability of F-type motoneurons was unchanged in the mutant mice. Surprisingly, the S-type motoneurons of mSDO1 mice did display intrinsic hyperexcitability (lower rheobase, hyperpolarized spiking threshold). As S-type motoneurons are resistant in ALS, we conclude that early intrinsic hyperexcitability does not contribute to motoneuron degeneration.

## Introduction

Glutamate excitotoxicity has long been suggested to contribute to the degeneration of motoneurons in amyotrophic lateral sclerosis. Intrinsic hyperexcitability of motoneurons, which increases discharge probability and thereby calcium inflow, has been assumed to participate in the excitotoxic process (*Ilieva et al., 2009*). However, it was recently suggested that hyperexcitability improves motoneuron survival (*Saxena et al., 2013*). Regardless of its effect, it is still not clear whether spinal motoneurons are hyperexcitable in mutant superoxide dismutase 1 (mSOD1) mice, a standard model of amyotrophic lateral sclerosis (ALS). Indeed, changes in excitability occur very early in mSOD1 mice (*Elbasiouny et al., 2010*). Motoneurons from mSOD1 embryos recorded in culture are hyperexcitable (*Pieri et al., 2003*; *Kuo et al., 2005*): they are recruited at lower current and display higher F–I gain. Similarly, *Martin et al. (2013)* found, in an in vitro preparation of mSOD1 embryonic cord, that motoneurons are also hyperexcitable: their dendritic tree is reduced causing an increase in input resistance. Investigations in neonates have led to contradictory results. Hypoglossal motoneurons were reported to be hyperexcitable (F–I gain is increased, *van Zundert et al., 2008*). However, *Pambo–Pambo et al. (2009)* did not observe any change in spinal motoneuron input resistance, rheobase, or stationary gain suggesting that their excitability was unchanged. In the same line, *Quinlan et al. (2011)* found that the excitability of spinal motoneurons is homeostatically maintained despite an increase in their input conductance (recruitment current and F–I gain unchanged). In contrast, *Bories et al. (2007)* reported a decrease in input resistance causing the spinal motoneurons to be hypoexcitable.

These discrepancies might be due to the location of the mutation on the SOD1 gene, the number of transgenes, or other factors. Until now, however, the fact that the motor unit population is

**\*For correspondence:** felxfel@aol.com

**Competing interests:** The authors declare that no competing interests exist.

**eLife digest** Amyotrophic lateral sclerosis (ALS), which is also known as Lou Gherig's disease or motoneuron disease, is a neurodegenerative disorder in which muscles throughout the body gradually waste away due to the death of the neurons that control their activity. The disease often begins with weakness of the arms or legs, but progresses to include difficulties with movements such as swallowing and breathing. Around half of those affected die within 3 or 4 years of diagnosis.

Although the causes of the disease are unclear, one leading theory is that the neurons that control muscle activity—motoneurons—are hyperexcitable during early development, and therefore fire too frequently. This causes too much calcium to enter the neurons and, because calcium is toxic to cells in high quantities, leads ultimately to the death of the neurons. But despite the popularity of this idea, and the fact that many therapeutic assays for ALS are based on attempts to reverse this process, there is no direct evidence that early hyperexcitability of motoneurons causes their death in ALS.

Leroy et al. have now tested this theory directly by taking advantage of the fact that not all motoneurons are affected by ALS. The large 'F-type' motoneurons that control fast-contracting muscle fibres degenerate in ALS, whereas the small 'S-type' motoneurons that control slow-contracting muscle fibres do not. A comparison of F-type and S-type motoneurons in a mouse model of ALS revealed that, surprisingly, S-type motoneurons are hyperexcitable in young ALS mice, whereas F-type motoneurons are not.

Given that S-type motoneurons are resistant to the effects of ALS, this indicates that early hyperexcitability cannot be the cause of motoneuron degeneration. Previous studies have tended to pool different types of motoneurons together, which might explain why this difference has not been seen before. Further experiments are now required to determine whether the hyperexcitability of S-type motoneurons persists into adulthood, and whether it might even contribute to their survival in ALS.

heterogeneous, even in neonates (*Jansen and Flatby, 1990*), has never been taken into account. Motoneurons innervate different types of muscle fibers and display different patterns of discharge during the second post-natal week both in rats (*Russier et al., 2003*) and in mice (*Pambo–Pambo et al., 2009*). Indeed, for liminal current pulses, the discharge starts at the pulse onset in some motoneurons (*immediate firing* pattern) but is delayed in others (*delayed firing* pattern). Here, we provide electrical, morphological and molecular evidence that immediate firing spinal motoneurons innervate slow-contracting fibers (S-type motoneurons) whereas delayed firing motoneurons innervate fast-contracting fibers (F-type motoneurons). We then investigated whether these two populations are equally affected in neonatal mSOD1 mice. We show that this is not the case: only the immediate firing motoneurons are hyperexcitable. Their rheobase is decreased because of a more hyperpolarized voltage threshold for spiking. In sharp contrast, the excitability of the delayed firing motoneurons is unchanged. Since the F-type motoneurons are vulnerable in ALS whereas the S-type motoneurons are resistant (*Pun et al., 2006*; *Hegedus et al., 2008*), the remarkably selective intrinsic hyperexcitality of S-type motoneurons in neonates indicates that intrinsic hyperexcitability is not an early event that triggers degeneration of the motoneurons.

## Results

### Delayed and immediate firing motoneurons

In neonatal mice, spinal motoneurons can be sorted according to their firing pattern. *Figure 1A* illustrates how two motoneurons in a wild-type (WT) mouse discharge in response to a long (5 s) square pulse at rheobase, that is, the minimal current pulse that elicits at least one action potential in our protocol (see 'Materials and methods'). In the example of *Figure 1A1*, the motoneuron did not fire at the onset of a 1.6 nA square pulse. Instead the motoneuron depolarized slowly and started to fire only 2.9 s after the pulse onset when the membrane potential reached the voltage threshold for spiking ($-35$ mV, dashed line). Once the firing started, its frequency increased. This motoneuron

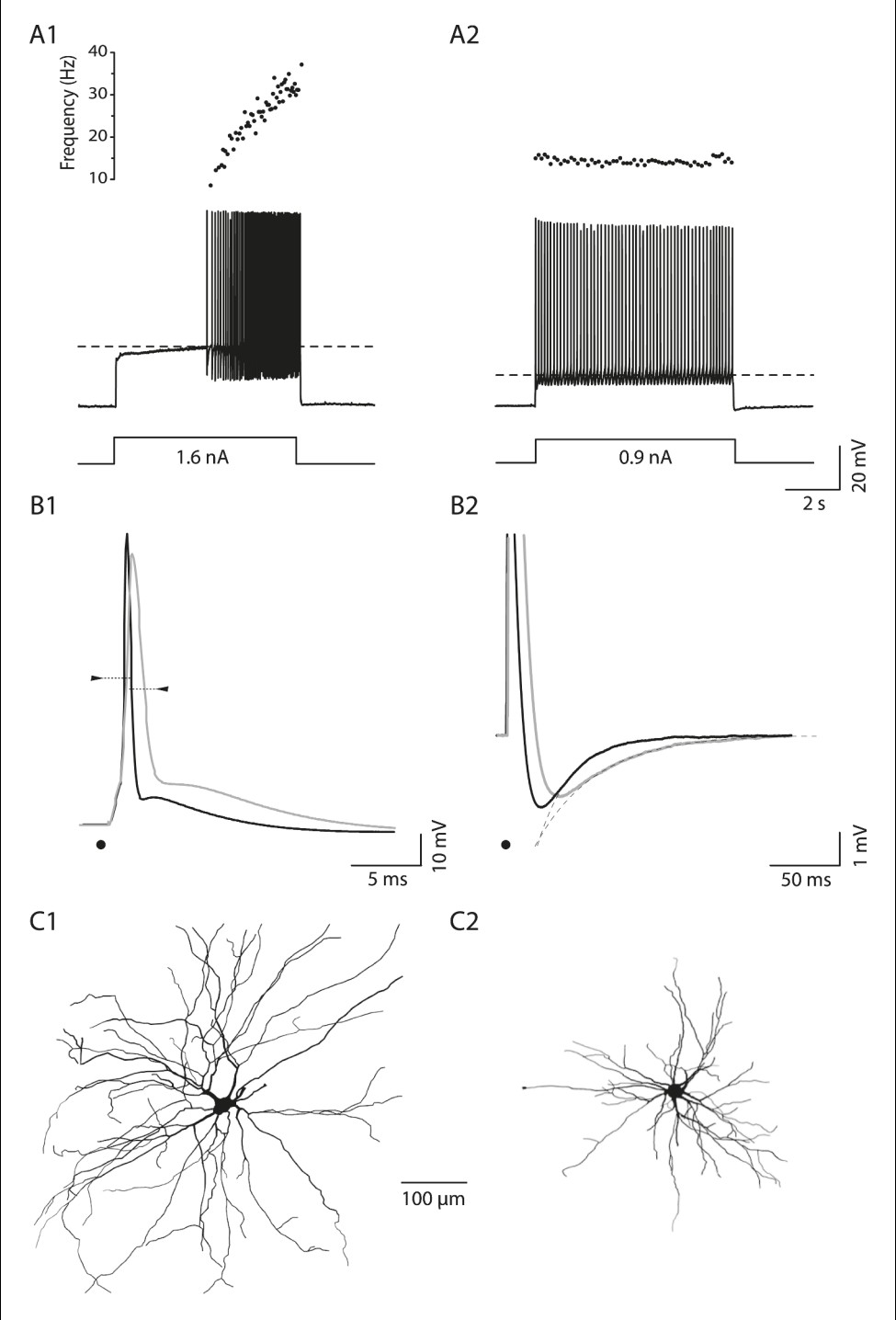

**Figure 1.** Electrical and morphological properties of motoneurons displaying the delayed and the immediate firing patterns. (**A1**) WT motoneuron displaying the delayed firing pattern in response to a 5 s pulse. The current intensity was the minimal intensity necessary to elicit firing in our searching protocol (rheobase). Bottom: injected-current (square pulses), middle: voltage-response and top: instantaneous firing frequency. The horizontal dashed line shows the voltage threshold for spiking (−50 mV). (**A2**) Response of a WT motoneuron displaying the immediate firing pattern. Same arrangement as in **A1**. (Voltage threshold for spiking: −59 mV). (**B1**) Single action potentials from a delayed (black line) and an immediate (gray line) firing WT motoneurons elicited by a short square pulse of current. The arrowheads point to the horizontal dotted bar drawn at half action potential amplitudes. (**B2**) Same records at a longer time base in order to show the after hyperpolarisation (AHP). Dashed

*Figure 1 continued on next page*

*Figure 1 continued*

lines are the exponential fits of the AHP relaxation. The relaxation time constants are 17 ms and 37 ms for the delayed firing and the immediate firing motoneurons, respectively. Note that the relaxation time constant is longer in the immediate firing motoneuron than in the delayed firing one. **C** Reconstructed dendritic trees of WT delayed (**C1**) and immediate (**C2**) firing motoneurons. The axon was not reconstructed in either case. Reconstructions are projected in the same plane as the slice.

The following figure supplement is available for figure 1:

**Figure supplement 1.** In delayed firing motoneurons, the delay depends on the intensity of stimulation.

displayed the *delayed firing* pattern. A very important feature of this pattern is illustrated in *Figure 1—figure supplement 1*. The delay was long at rheobase (*Figure 1—figure supplement 1A*) but it progressively decreased (*Figure 1—figure supplement 1B–C*) to finally disappear (*Figure 1—figure supplement 1D*) as the current intensity increased. 63 out of 94 WT motoneurons (67%) exhibited this *delayed firing* pattern. The remaining motoneurons displayed the so-called *immediate firing* pattern: at rheobase, the motoneuron discharged at the pulse onset without any delay (*Figure 1A2*). In the immediate firing pattern, the spiking frequency remained constant with little variability. 31 out of the 94 WT motoneurons (i.e., 33%) displayed the immediate firing pattern.

On average, the input conductance is smaller, the rheobase is lower and the voltage threshold for spiking is more hyperpolarized in immediate firing motoneurons than in delayed firing ones and this occurs despite of the fact that the resting membrane potential is similar (*Table 1*). Moreover, immediate and delayed firing motoneurons display differences in the shape of their action potentials. Immediate firing motoneurons have wider action potentials and a longer relaxation time constant of their after-hyperpolarization (AHP) compared to delayed firing motoneurons (see *Figure 1B* and *Table 1*). These results suggest that the two phenotypes (immediate and delayed firing) are linked to two different populations of motoneurons.

## Delayed and immediate firing motoneurons display different morphologies

Further supporting the hypothesis of two separate populations, we found that the dendritic trees of delayed and immediate firing motoneurons display different morphologies. We filled motoneurons with neurobiotin in order to investigate the dendritic tree architecture. Only the dendrites that remained in the slice plane were considered for analysis (see 'Materials and methods'). In these conditions, the number of primary dendrites (and thereby dendritic trees) per motoneuron is similar in immediate and delayed firing motoneurons (*Table 2*). This allows us to make relevant morphological comparisons on the reconstructed trees. As exemplified in *Figure 1C*, the dendritic arborization extends further in delayed firing motoneurons than in immediate firing motoneurons (compare *Figure 1C1* and *Figure 1C2*). In WT mice, delayed firing motoneurons have on average more branching points (44 ± 14, 26 to 72, N = 14) compared to the immediate firing motoneurons (27 ± 13, 13 to 52, N = 10, p = 0.007), larger total dendritic length and longer dendritic paths than immediate firing motoneurons (*Table 2*).

## Both immediate and delayed firing motoneurons are alpha-motoneurons

A body of evidence indicates that both immediate and delayed firing motoneurons are alpha-motoneurons (i.e., motoneurons that innervate extrafusal muscle fibers) and not gamma-motoneurons (i.e., motoneurons that innervate intrafusal muscle fibers). First, the soma sizes (*Table 2*) of both immediate and delayed firing motoneurons are in the range of alpha motoneurons (see supplemental data in *Friese et al. (2009)* for the soma size distribution of alpha and gamma-motoneurons in P14 mice). Note, however, that on average, immediate firing motoneurons have smaller soma areas than delayed firing motoneurons (*Table 2*). Second, since alpha-motoneurons, but not gamma-motoneurons, receive monosynaptic Ia inputs (*Eccles et al., 1960*), we checked whether immediate and delayed firing motoneurons receive such proprioceptive inputs. We recorded motoneurons, characterized their discharge pattern and filled them with neurobiotin. Subsequently, we stained the

**Table 1.** Electrophysiological properties

| | | WT mice | mSOD1 mice | p-value |
|---|---|---|---|---|
| Resting membrane potential (mV) | Delayed firing | −64 ± 3 | −65 ± 3 | 0.2 |
| | | −70/−56 | −70/−59 | |
| | | N = 63 | N = 31 | |
| | Immediate firing | −65 ± 3 | −64 ± 2 | 0.2 |
| | | −71/−59 | −70/−60 | |
| | | N = 31 | N = 18 | |
| | p-value | 0.3 | 0.1 | |
| Input conductance (nS) | Delayed firing | 52 ± 28 | 54 ± 30 | 0.8 |
| | | 10/151 | 22/153 | |
| | | N = 63 | N = 31 | |
| | Immediate firing | 33 ± 24 | 33 ± 16 | 0.6 |
| | | 6/98 | 6/62 | |
| | | N = 31 | N = 18 | |
| | p-value | 0.0007 | 0.01 | |
| Rheobase (nA) | Delayed firing | 1.2 ± 0.6 | 1.1 ± 0.5 | 0.4 |
| | | 0.3/2.8 | 0.3/2.6 | |
| | | N = 57 | N = 30 | |
| | Immediate firing | 0.6 ± 0.4 | 0.3 ± 0.2 | 0.008 |
| | | 0.05/1.6 | 0.1/0.6 | |
| | | N = 29 | N = 16 | |
| | p-value | <0.0001 | <0.0001 | |
| Voltage threshold for spiking (mV) | Delayed firing | −33 ± 7 | −31 ± 10 | 0.7 |
| | | −47/−17 | −50/−10 | |
| | | N = 58 | N = 30 | |
| | Immediate firing | −44 ± 7 | −49 ± 6 | 0.03 |
| | | −50/−41 | −50/−30 | |
| | | N = 30 | N = 17 | |
| | p-value | <0.0001 | <0.0001 | |
| Voltage threshold for spiking–Resting membrane potential (mV) | Delayed firing | 31 ± 8 | 33 ± 10 | 0.4 |
| | | 17/49 | 13/50 | |
| | | N = 59 | N = 31 | |
| | Immediate firing | 20 ± 7 | 14 ± 5 | 0.005 |
| | | 8/31 | 6/21 | |
| | | N = 30 | N = 17 | |
| | p-value | <0.0001 | <0.0001 | |
| Recruitment current on ramp (nA) | Delayed firing | 1.1 ± 0.6 | 1.1 ± 0.5 | 0.8 |
| | | 0.1/2.8 | 0.3/2.5 | |
| | | N = 51 | N = 29 | |
| | Immediate firing | 0.6 ± 0.5 | 0.3 ± 0.3 | 0.02 |
| | | 0.07/2 | 0.07/1 | |
| | | N = 25 | N = 15 | |
| | p-value | 0.001 | <0.0001 | |

*Table 1 continued on next page*

*Table 1 continued*

|  |  | WT mice | mSOD1 mice | p-value |
|---|---|---|---|---|
| Action potential amplitude (mV) | *Delayed firing* | 89 ± 13 | 87 ± 11 | *0.5* |
|  |  | 66/121 | 71/111 |  |
|  |  | N = 29 | N = 19 |  |
|  | *Immediate firing* | 84 ± 11 | 81 ± 15 | *0.4* |
|  |  | 66/104 | 61/110 |  |
|  |  | N = 21 | N = 13 |  |
|  | *p-value* | *0.2* | *0.2* |  |
| Action potential width (ms) | *Delayed firing* | 1.4 ± 0.5 | 1.3 ± 0.4 | *0.6* |
|  |  | 0.7/2.5 | 0.6/2.2 |  |
|  |  | N = 29 | N = 19 |  |
|  | *Immediate firing* | 1.7 ± 0.4 | 1.8 ± 0.6 | *0.7* |
|  |  | 1.1/2.9 | 0.9/3.1 |  |
|  |  | N = 21 | N = 13 |  |
|  | *p-value* | *0.04* | *0.01* |  |
| AHP relaxation time constant (ms) | *Delayed firing* | 27 ± 9 | 23 ± 5 | *0.2* |
|  |  | 11/50 | 15/34 |  |
|  |  | N = 21 | N = 12 |  |
|  | *Immediate firing* | 42 ± 12 | 48 ± 27 | *1* |
|  |  | 21/60 | 19/91 |  |
|  |  | N = 11 | N = 7 |  |
|  | *p-value* | *0.004* | *0.02* |  |

vesicular glutamate transporter 1 (VGlut1) since VGlut1 is expressed in terminals from primary afferents but not in terminals from excitatory interneurons or descending fibers (*Oliveira et al., 2003*; *Friese et al., 2009*). VGlut1 afferents are known to innervate not only alpha-motoneurons but also Renshaw cells in the neonate (*Mentis et al., 2006*). However, since we identified the labelled cells as motoneurons on the basis of antidromic action potential recordings, we are confident that VGlut1 terminals shown on *Figure 2A* are apposed on alpha-motoneurons and not on Renshaw cells. We observed that both delayed and immediate firing motoneurons received proprioceptive terminals on the soma and proximal dendrites (*Figure 2A*, arrows in insets). Third, we checked for the expression of the neuronal nuclear antigen (NeuN), a known marker for alpha-motoneurons (*Friese et al., 2009*; *Shneider et al., 2009*). Both immediate and delayed firing motoneurons expressed NeuN (*Figure 2B*).

## Molecular markers suggest that immediate firing motoneuron are S-type motoneurons whereas delayed firing motoneurons are F-type motoneurons

It has been suggested that the estrogen-related receptor β (Errβ) is expressed in S-type but not in F-type motoneurons. Conversely, chondrolectin (Chodl) is expressed in a fraction of F-type motoneurons but not in S-type motoneurons (*Enjin et al., 2010*). We therefore tested the expression of these two molecular markers on immediate and delayed firing motoneurons. *Figure 3A* shows an immediate firing motoneuron expressing Errβ and a delayed firing motoneuron that did not. Remarkably, all 6 investigated immediate firing motoneurons proved to be Errβ-positive whereas all 8 delayed firing motoneurons were Errβ-negative. This molecular distinction matches with differences in the electrical properties (*Figure 3B*). In situ hybridization allowed us to investigate whether immediate and delayed firing motoneurons expressed Chodl mRNA (*Figure 4A*). None of the 9 investigated immediate firing motoneurons did. 7 out of the 15 investigated delayed firing

**Table 2.** Morphological properties

| | | WT mice | mSOD1 mice | p-value |
|---|---|---|---|---|
| Soma area (µm$^2$) | Delayed firing | 630 ± 160 | 620 ± 140 | 0.2 |
| | | 350/1000 | 270/890 | |
| | | N = 60 | N = 31 | |
| | Immediate firing | 530 ± 180 | 454 ± 110 | 0.2 |
| | | 260/940 | 250/640 | |
| | | N = 30 | N = 17 | |
| | p-value | 0.009 | 0.0002 | |
| Primary dendrites | Delayed firing | 6.4 ± 2.0 | 6.7 ± 1.2 | 0.3 |
| | | 4/12 | 5/9 | |
| | | N = 14 | N = 14 | |
| | Immediate firing | 6.3 ± 2.7 | 6.4 ± 4.0 | 0.9 |
| | | 3/10 | 2/13 | |
| | | N = 10 | N = 5 | |
| | p-value | 1 | 0.5 | |
| Total dendritic length (mm) | Delayed firing | 8.3 ± 2.9 | 8.7 ± 3.8 | 0.9 |
| | | 2.3/14 | 3.8/16.5 | |
| | | N = 14 | N = 14 | |
| | Immediate firing | 5.3 ± 1.5 | 3.6 ± 0.3 | 0.01 |
| | | 3.0/8.0 | 3.3/4.1 | |
| | | N = 10 | N = 5 | |
| | p-value | 0.01 | 0.0003 | |
| Dendritic paths (µm) | Delayed firing | 296 ± 135 | 293 ± 139 | 0.8 |
| | | 5/687 | 15/840 | |
| | | N = 653 | N = 449 | |
| | Immediate firing | 252 ± 141 | 181 ± 157 | <0.0001 |
| | | 11/803 | 11/685 | |
| | | N = 281 | N = 180 | |
| | p-value | <0.0001 | <0.0001 | |
| Terminal segments length (µm) | Delayed firing | 112 ± 93 | 108 ± 102 | 0.5 |
| | | 4/457 | 3/742 | |
| | | N = 618 | N = 447 | |
| | Immediate firing | 108 ± 91 | 81 ± 92 | 0.001 |
| | | 5/545 | 4/584 | |
| | | N = 296 | N = 178 | |
| | p-value | 0.6 | 0.001 | |

Note that the 'overbranching motoneuron' (arrowhead on **Figure 8B1**) is excluded for analysis.

motoneurons were Chodl-positive whereas the remaining 8 were Chodl-negative. Altogether, the immediate firing motoneurons were all Errβ-positive and Chodl-negative as expected for S-type. On the other hand, the delayed firing motoneurons were all Errβ-negative and about half of them were Chodl-positive whereas the other half were chondrolectine-negative as expected for F-type motoneurons (**Enjin et al., 2010**). Interestingly, the Chodl-positive motoneurons tended to display the highest rheobases (**Figure 4B**).

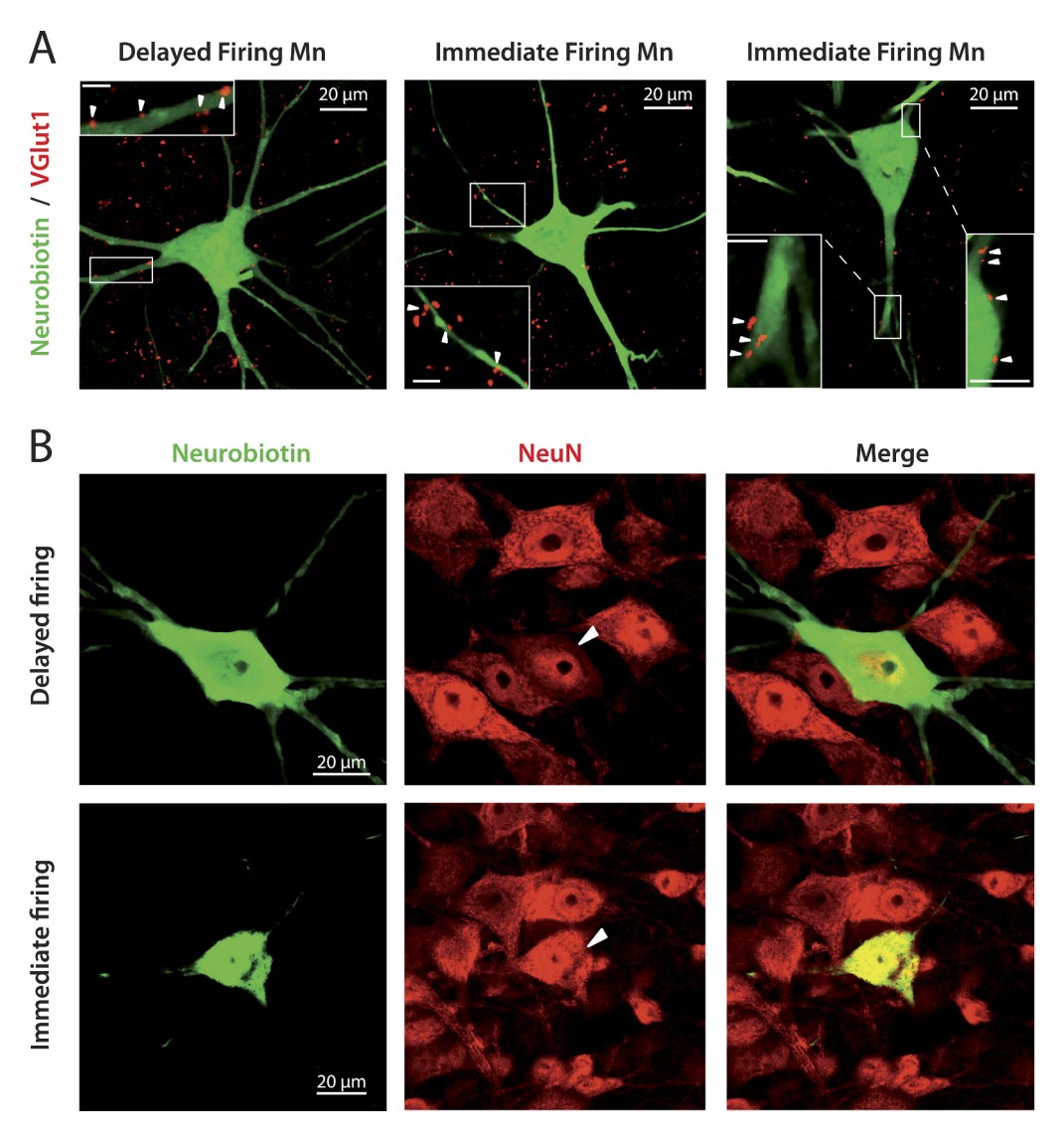

**Figure 2.** immediate and delayed firing motoneurons both receive VGlut1 inputs and express NeuN. (**A**) Vglut1 (red) synaptic inputs are apposed to neurobiotin (green) filled motoneurons (examples of appositions pointed by arrowheads in the inserts that show enlargements of the areas surrounded by rectangles). The bar scale in all insets is 5 μm. (**B**) NeuN staining (red) of neurobiotin (green) filled motoneurons. The arrowheads point to the cell bodies of the motoneurons that have been intracellularly filled during the electrophysiological experiment.

## MMP9 expression profile indicates that immediate firing motoneurons are resistant in ALS

In addition to identifying them as F- and S-type motoneurons, we set out to directly identify which motoneurons sub-population was vulnerable in ALS. *Kaplan et al. (2014)* recently showed that the matrix metalloproteinase-9 (MMP9) is strongly expressed in motoneurons that are the most vulnerable to ALS, that is, those of the large, fast contracting and fatigable (FF) motor units. Conversely, the most resistant motoneurons, that is, those that innervate the slow contracting fibers, were devoid of MMP9. We therefore investigated whether MMP9 was expressed in delayed and immediate firing motoneurons. All 5 investigated immediate firing motoneurons were MMP9 negative (*Figure 5A*, first row). In contrast, 5 out of the 10 investigated delayed firing motoneurons displayed a strong MMP9 labelling (*Figure 5A*, second row). Four other delayed firing motoneurons did not express

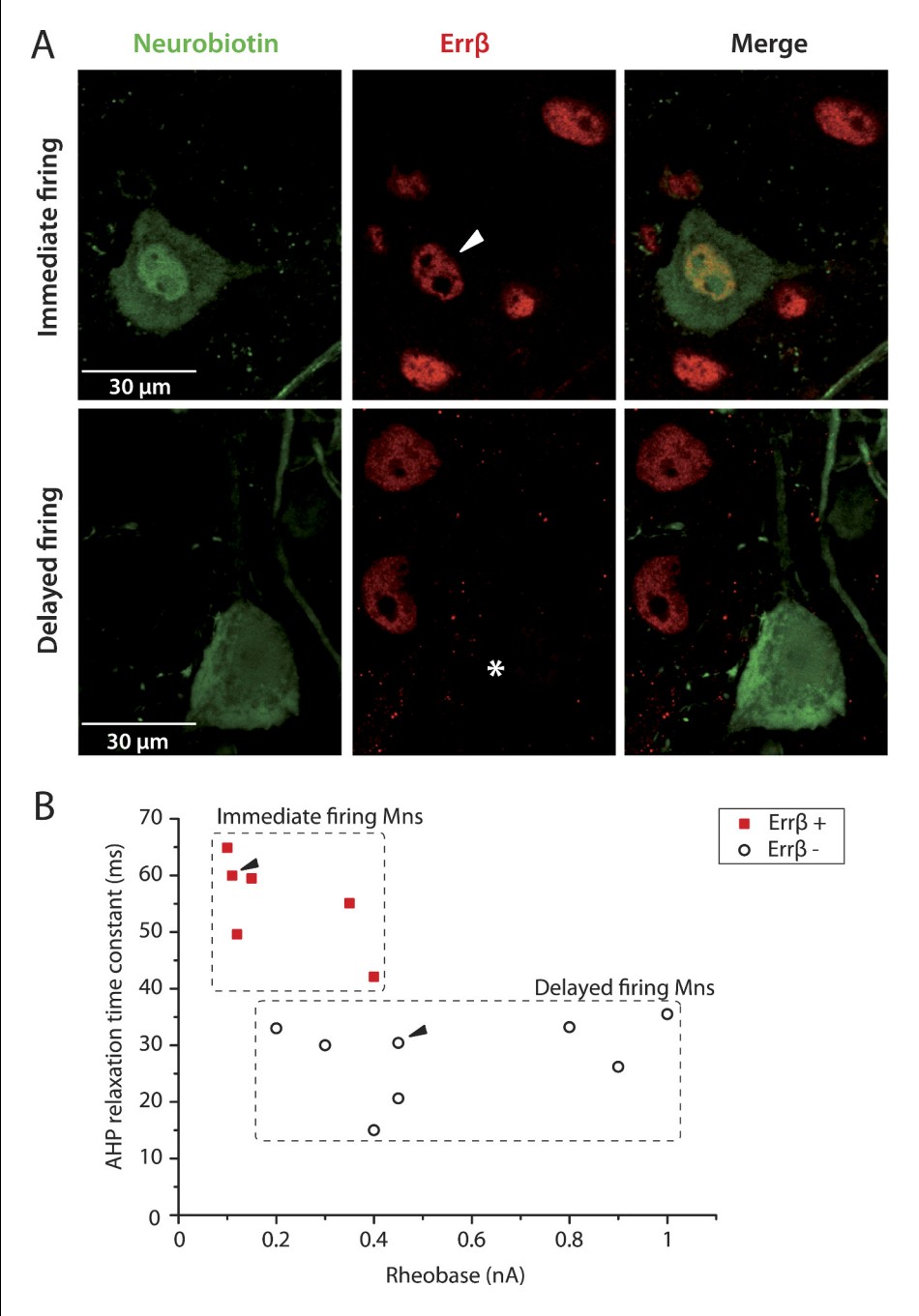

**Figure 3.** Immediate firing motoneurons, but not delayed firing motoneurons, express ERRβ. (**A**) Examples of Errβ. staining (red) in neurobiotin (green) filled motoneurons. The arrowhead in the first row points to the nucleus (Errβ−positive) of the recorded motoneuron. The asterisk in the second row indicates that the nucleus of the recorded motoneuron was Errβ−negative. (**B**) Plot of the AHP relaxation time constants against the rheobases for labelled motoneurons. All immediate firing motoneurons were Errβ−positive whereas all delayed firing motoneurons were Errβ−negative. Arrowheads point to the motoneurons illustrated in **A**.

MMP9 (*Figure 5A*, third row) and one was weakly labelled. The delayed firing motoneurons that express MMP9 tend to exhibit the largest input conductances and rheobases (*Figure 5B*) suggesting that they innervate the largest motor units. These results indicate that the immediate firing moto-neurons are resistant during ALS, in keeping with our identification as S-type motoneurons.

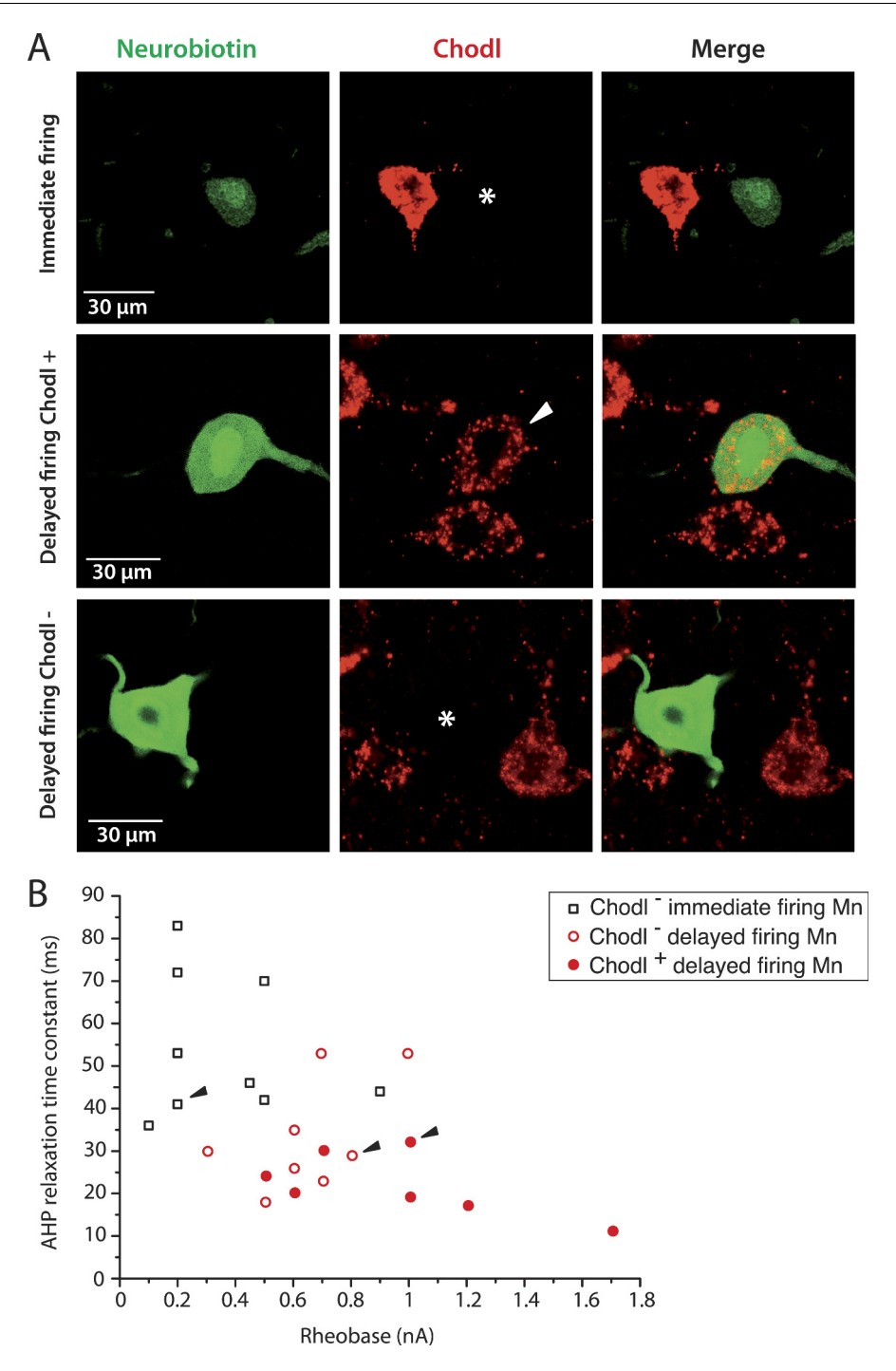

**Figure 4.** The largest delayed firing motoneurons express chondrolectin mRNA contrary to immediate firing motoneurons. (**A**) Examples of chondrolectin in situ hybridizations (red) in neurobiotin filled motoneurons (green). The asterisks in the first and third rows indicate that the cell body of the recorded motoneuron was chondrolectin negative. The arrowhead in the second row indicates that chondrolectin is expressed in this delayed firing motoneuron. (**B**) Plot of the AHP relaxation time constants against the rheobase for the investigated motoneurons. About half of the delayed firing motoneurons (the ones that display the highest rheobase) were chondrolectin-positive. All the immediate firing motoneurons were chondrolectin-negative. Arrowheads point to the motoneurons illustrated in **A**.

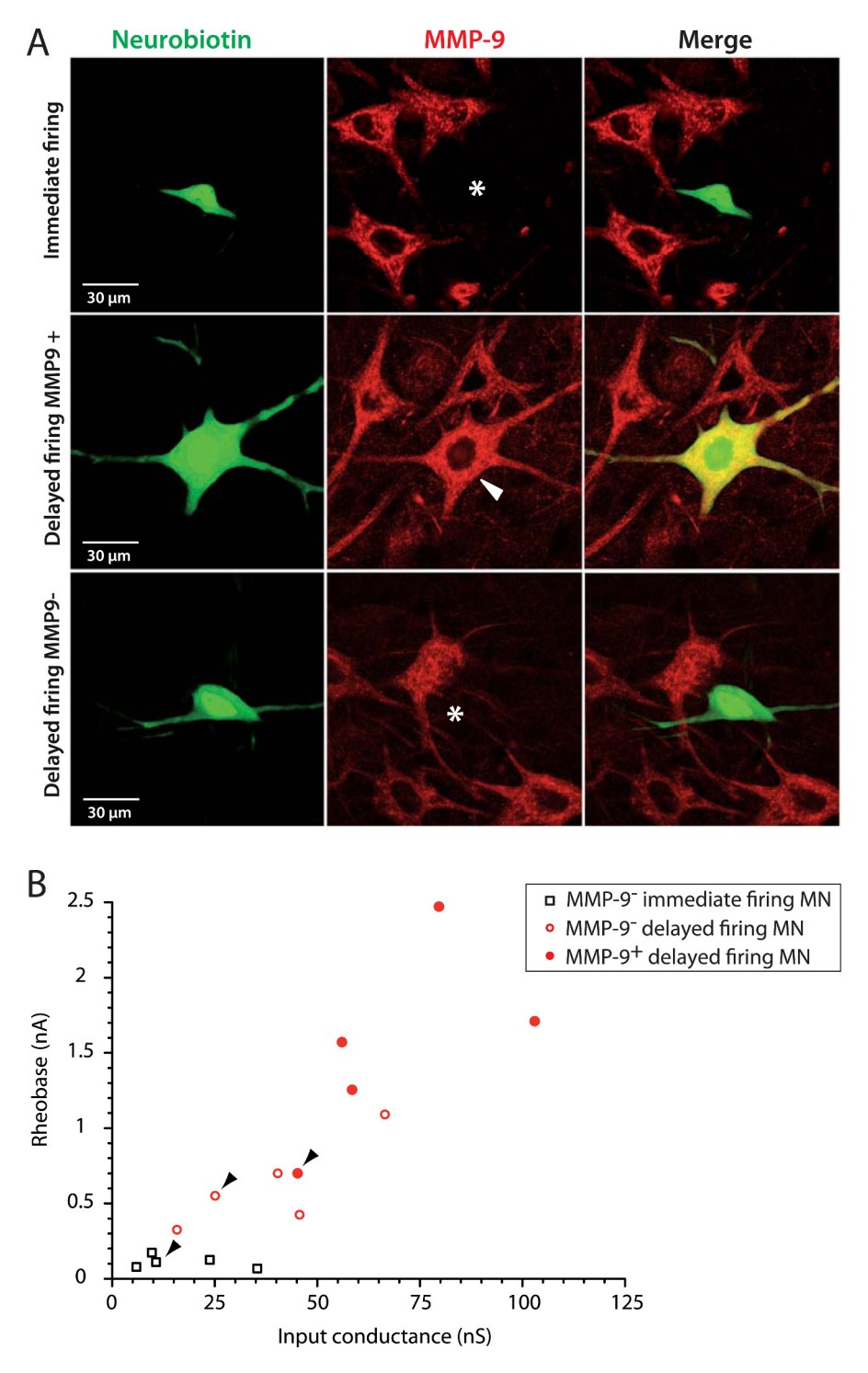

**Figure 5.** The largest delayed firing motoneurons express MMP9 contrary to immediate firing motoneurons. (**A**) Examples of MMP9 labelling in neurobiotin filled motoneurons (green). The asterisks in the first and third rows indicate the cell body of the recorded motoneurons devoid of MMP9 expression. The arrowhead in the second row indicates that MMP9 was expressed in this delayed firing motoneuron. (**B**) Plot of the rheobase against the input conductance for the investigated motoneurons. Half of the delayed firing motoneurons (the ones that display the highest rheobases and input conductances) were MMP9-positive. All the immediate firing motoneurons were MMP9-negative. Arrowheads point to the motoneurons illustrated in **A**.

## Excitability of the immediate firing motoneurons is selectively increased in mSOD1 mice

In mSOD1 mice, we observed the same immediate and delayed firing patterns (*Figure 6A*) as in WT mice. 31 out of 49 mSOD1 motoneurons (63%) exhibited the *delayed firing* pattern (*Figure 6A1*) whereas the 18 remaining motoneurons (37%) displayed the *immediate firing* pattern (*Figure 6A2*). The proportion of delayed and immediate firing motoneurons is not different between WT and mSOD1 mice (*Fisher's exact* test, p = 0.5). Moreover, the resting membrane potential, the input conductance, the action potential width and the AHP relaxation time constant of each motoneuron subtype are unchanged in mSOD1 mice compared to WT mice (*Table 1*).

Despite the unchanged input conductance, the excitability of the immediate firing motoneurons, but not of the delayed firing ones, is altered in mSOD1 mice. As expected, the rheobase, that is, a measure of cell excitability, increases with input conductance (*Figure 6B*). However, in immediate

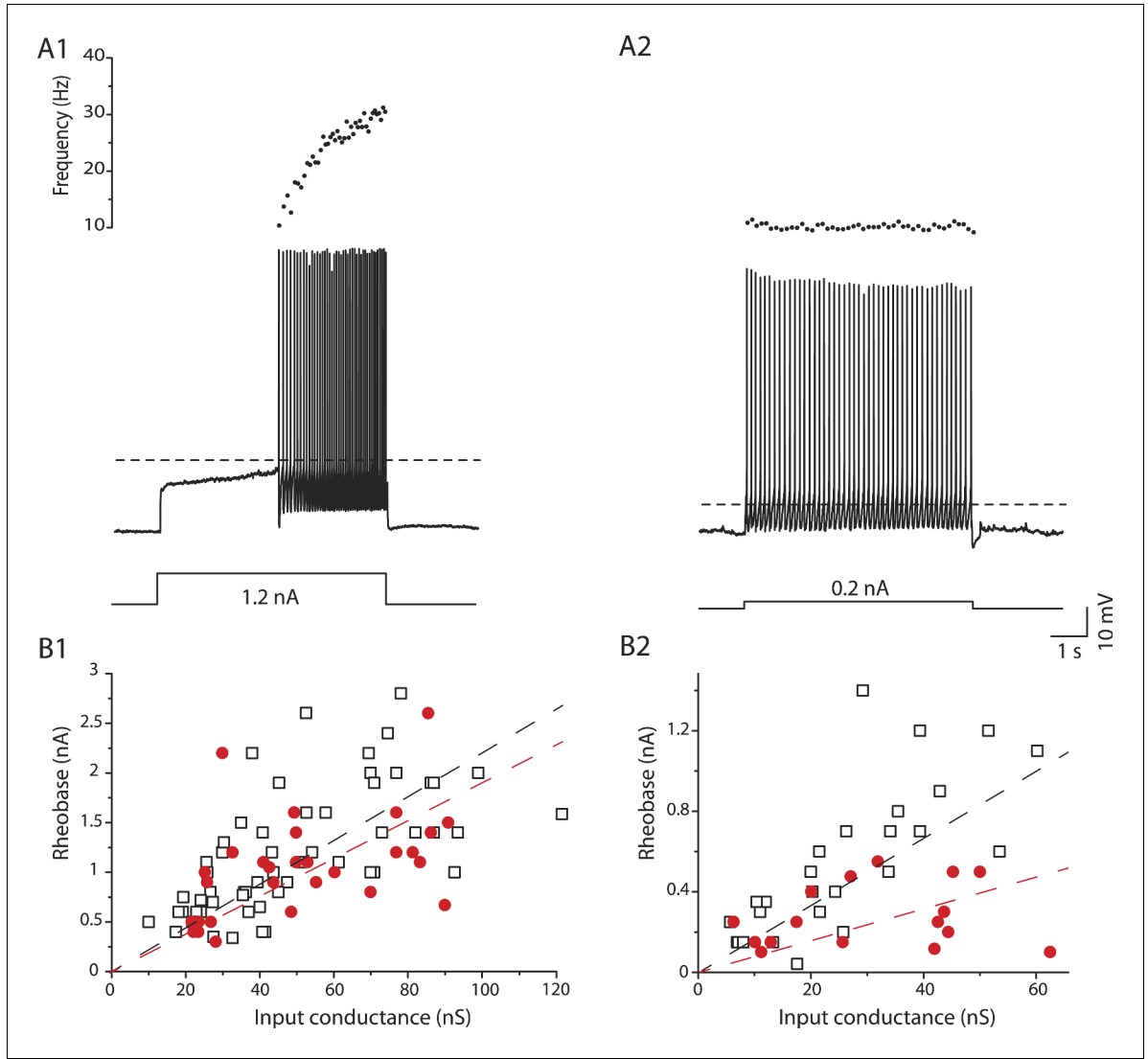

**Figure 6.** mSOD1 immediate firing motoneurons are selectively hyperexcitable. (A1-2) mSOD1 motoneurons displaying the delayed firing pattern (A1) and the immediate firing pattern (A2). The current intensity was the minimal intensity necessary to elicit firing (rheobase). Bottom: injected-current (square pulses), middle: voltage-response and top: instantaneous firing frequency. The horizontal dashed line shows the voltage threshold for spiking (−37 mV for the delayed firing motoneuron and −48 mV for the immediate firing motoneuron). (B1-2) Plot of rheobase as a function of input conductance for delayed (B1) and immediate (B2) firing motoneurons. In each plot, open squares are for WT motoneurons whereas red dots are for mSOD1 motoneurons. Linear regressions are indicated by dashed lines.

firing motoneurons, the slope of the linear regressions is significantly smaller in mSOD1 mice than in WT mice (*Figure 6B2*, 15 vs 22 mV *t* test p = 0.04, Chow test p = 0.0004). This is not the case for delayed firing motoneurons (*Figure 6B1*, 19 vs 22 mV, *t* test p = 0.5, Chow test p = 0.2). On average the rheobase of immediate firing motoneurons is two times smaller in mSOD1 mice compared to WT mice (*Table 1*). On the other hand, the rheobase of delayed firing motoneurons is not significantly affected in mSOD1 mice (*Table 1*). However, the resting membrane potential of immediate firing motoneurons is unchanged in mSOD1 mice (*Table 1*). The decrease in rheobase in mSOD1 immediate firing motoneurons is instead due to an hyperpolarization of the voltage threshold for spiking (*Table 1*). Accordingly, the difference between the voltage threshold for spiking and the resting membrane potential is smaller for the immediate firing motoneurons in mSOD1 mice (*Table 1*). As a consequence, a smaller amount of current is required to reach the voltage threshold for spiking in these motoneurons. Motoneuron excitability was also assessed on the basis of their responses to slow triangular ramps of current (*Figure 7—figure supplement 1*). The current at which the first action potential was fired (recruitment current) during a slow ramp is indeed another way to measure the rheobase. The recruitment current on the slow ramps was very close to the rheobase measured using the current pulses (*Table 1*), and again, it was significantly smaller in mSOD1 mice only in the immediate firing motoneurons. Regardless of the way we measured the rheobase (long pulses or slow ramps) we found that the immediate firing motoneurons, but not the delayed firing ones, are hyperexcitable in mSOD1 mice.

We have previously shown that mixed mode oscillations (MMOs) are related to the excitability state of motoneurons (*Iglesias et al., 2011*). MMOs are small oscillations of the membrane potentials between full action potentials (arrowheads in *Figure 7A2*) (*Manuel et al., 2009*). They create variability in the firing discharge. In most cases they are present only at low current intensity, defining a sub-primary firing range (*Manuel et al., 2009*; *Turkin et al., 2010*). We have shown in a previous study (*Iglesias et al., 2011*) that MMOs are caused by a relative deficit of sodium current with

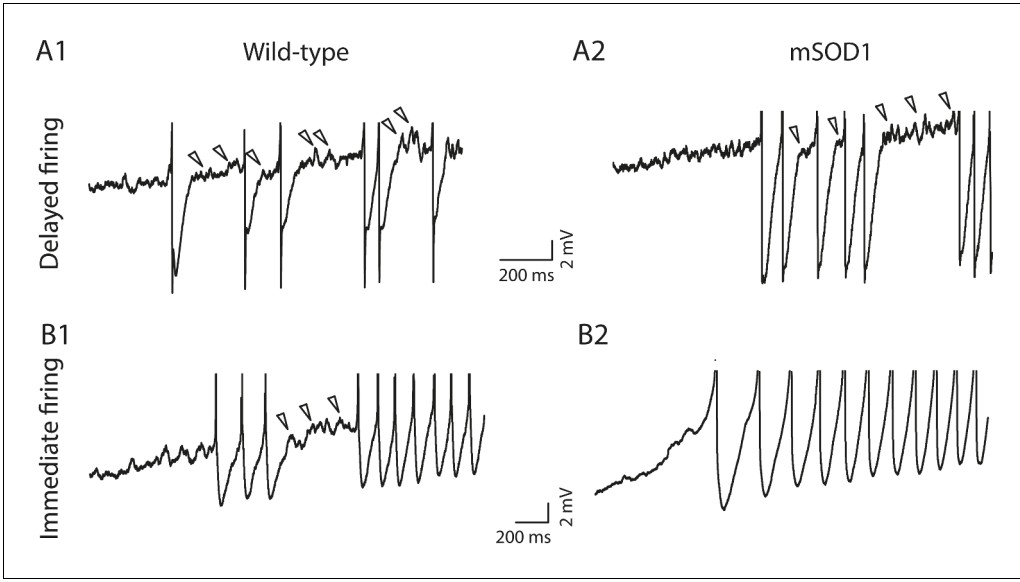

**Figure 7.** Absence of mixed mode oscillations in mSOD1 immediate firing motoneurons. Beginning of the discharge during the injection of slow triangular ramp of current at 0.1 nA/s velocity. See *Figure 7—figure supplement 1* for the full traces and F–I curves. Arrowheads point to oscillations between full spikes, the signature of mixed mode oscillations. Note that there is no oscillations between spikes in the mSOD1 immediate firing motoneuron (**B2**).

The following figure supplement is available for figure 7:

**Figure supplement 1.** Responses of a delayed firing motoneuron (upper row) and of an immediate firing motoneuron (lower row) to a slow current ramp.

respect to potassium current, which is due to a slow sodium inactivation. MMOs therefore reflect a low excitability state. In the case of the delayed firing motoneurons, MMOs were observed nearly in all cells (49 out of 50) recorded in WT animals (*Figure 7A1*) and in all 31 mSOD1 motoneurons (*Figure 7A2*). This is again an indication that their exitability is unaltered by the mutation. In the case of the immediate firing motoneurons, MMOs are encountered in nearly all WT motoneurons (13 out of 15, *Figure 7B1*) but they are absent in most mSOD1 motoneurons (MMOs are lacking in 8 out of 11 mSOD1 immediate firing motoneurons, *Figure 7B2*). The proportion of immediate firing motoneurons exhibiting MMOs in mSOD1 is thus significantly reduced compared to WT motoneurons (Fisher's exact test, p = 0.003). The absence of MMOs in the immediate firing motoneurons of mSOD1 mice further indicates that these motoneurons are more excitable than in WT mice.

When a primary range could be observed in immediate firing motoneurons (*Figure 7—figure supplement 1C2,D2*, dark points), its slope was not significantly different between WT and mSOD1 mice (35 ± 22 Hz nA$^{-1}$, 11–80 Hz nA$^{-1}$, N = 12 for WT vs 26 ± 8 Hz nA$^{-1}$, 16–41 Hz nA$^{-1}$ for mSOD1, N = 9, p = 0.7). Since the gain in the primary range is largely determined by the AHP conductance (*Brownstone et al., 1992*; *Kernell, 2006*; *Manuel et al., 2006*), this result suggests that the AHP conductance is unchanged in mSOD1 mice. In delayed firing motoneurons, a gain could not be measured because it is difficult to identify a linear primary range (see *Figure 7—figure supplement 1A2,B2*, dark points). However, we measured the firing frequency reached 0.5 s after the recruitment and we found that it is not different between delayed firing motoneurons of WT (30 ± 7 Hz, 17 to 49 Hz, N = 42) and mSOD1 mice (29 ± 6 Hz, 13 to 42 Hz, N = 26).

## The SOD1 mutation affects selectively the dendritic tree of immediate firing motoneurons

In mSOD1 mice, similarly to WT mice, delayed firing motoneurons are larger than immediate firing ones (*Figure 8A*). However the morphology of the dendritic tree is affected specifically in immediate firing motoneurons. *Figure 8B* shows the relationships between the number of branching points and the total dendritic length for WT and mSOD1 mice: the more branching points (and therefore branches), the longer the total dendritic length. In the case of delayed firing motoneurons, WT and mSOD1 relationships are largely overlapping (*Figure 8B1*, slopes of the linear regressions: 0.18 vs 0.18 mm/branching point, *t* test p = 1, Chow test p = 0.7), except for one motoneuron (arrowhead in *Figure 8B1*) that displays more branching points and a longer total dendritic length than the largest WT motoneurons. This particular motoneuron thereby displays an overbranching of its dendritic tree. If we exclude this outlayer motoneuron, the total dendritic length, the length of dendritic paths (*Figure 8C1*) and the length of terminal segments are not different in delayed firing motoneurons of mSOD1 and WT mice (*Table 2*).

In sharp contrast, immediate firing motoneurons undergo profound changes in mSOD1 mice. Their total dendritic length is 32% smaller in mSOD1 motoneurons than in WT motoneurons (*Table 2*) while the number of branching points is not significatively different (mSOD1 mice: 27 ± 14, 9–45, N = 5; WT mice: 27 ± 13, 13–52, N = 10, p = 0.8). As a result the relationships between the total dendritic length and the number of branching points are very different (*Figure 8B2*, slopes of the linear regressions: 0.02 vs 0.1 mm/branching point *t* test p = 0.02, Chow test p = 0.009). Consistently, the dendritic paths and terminal segments are on average 28% and 25% shorter, respectively, in mSOD1 mice (*Table 2*). *Figure 8C2* shows that the distribution of the dendritic paths is strongly skewed towards small lengths in mSOD1 mice (Kolmogorov–Smirnov bilateral test, p < 0.0001). Altogether, our data indicate that the dendrites of immediate firing motoneurons are shorter in mSOD1 mice than in WT mice.

## Discussion

We show that only spinal motoneurons that display the immediate firing pattern undergo substantial electrical and morphological alterations in neonatal mSOD1 mice whereas no changes occur in the motoneurons displaying the delayed firing pattern. In immediate firing motoneurons, their dendrites are about 30% shorter in mSOD1 compared to WT mice. Furthermore their rheobase is decreased, their voltage threshold for spiking is hyperpolarized and mixed mode oscillations are largely absent. This indicates that the immediate firing motoneurons become more excitable in mSOD1 mice.

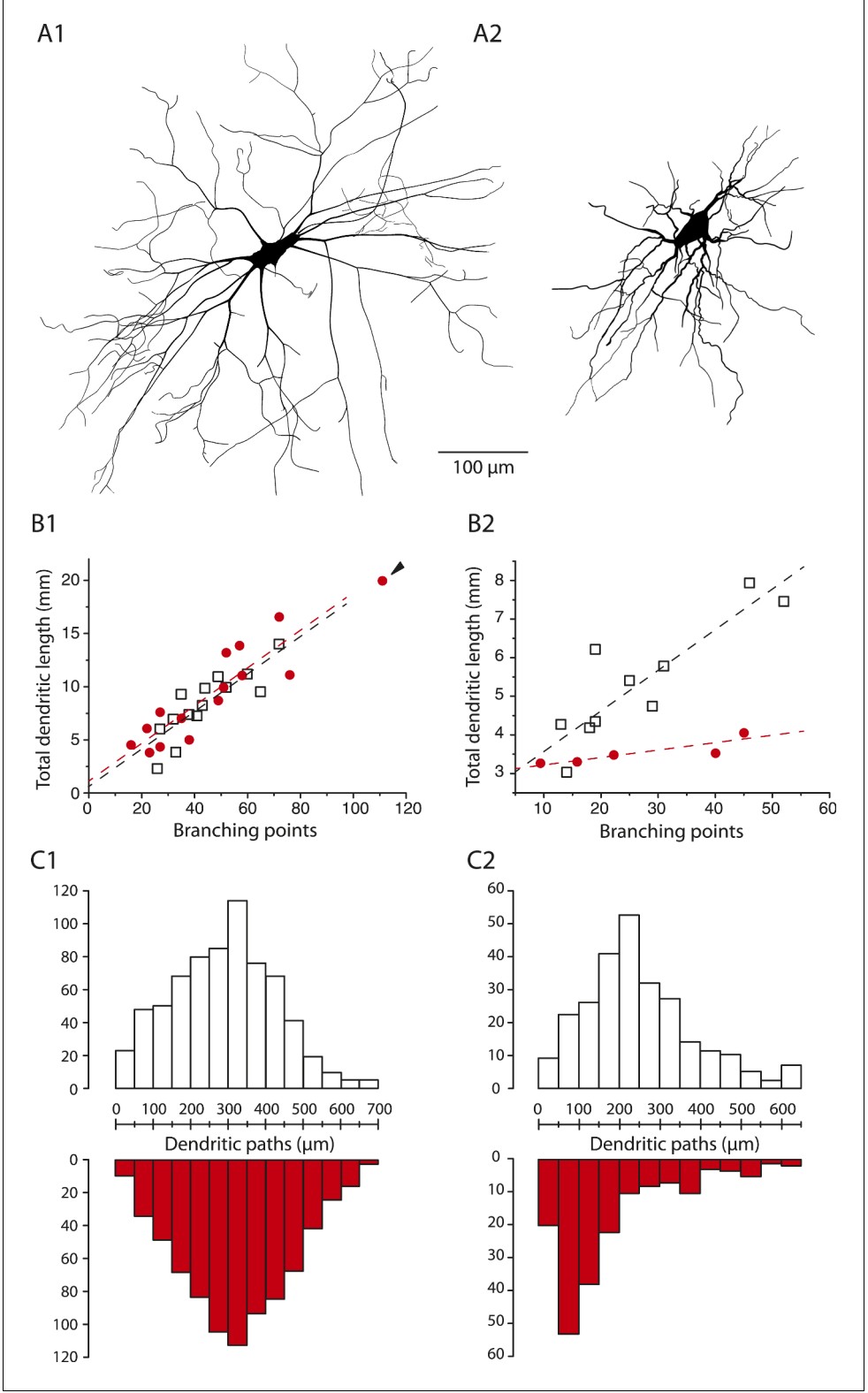

**Figure 8.** The dendritic tree of immediate firing motoneurons is shrunk in mSOD1 mice. (**A1-2**) Reconstructed dendritic trees of mSOD1 delayed (**A1**) and immediate (**A2**) firing motoneurons. The axons were not reconstructed. (**B1-2**) Plots of total dendritic length as a function of number of branching points for delayed firing motoneurons (**B1**) and immediate firing motoneurons (**B2**). In each plot, open squares are for WT motoneurons whereas red dots are for mSOD1 motoneurons. Arrowhead in **B1** points at an outlying over-branching mSOD1

*Figure 8 continued on next page*

*Figure 8 continued*

delayed motoneuron. Linear regressions are indicated by dashed lines. (**C1-2**) Distribution of the dendritic paths for delayed (**C1**) and immediate firing motoneurons (**C2**). Distributions between WT (open columns) and mSOD1 (red columns) dendritic paths are compared for each firing patterns.

## Is the discharge pattern related to specific motoneuron types?

We provide electrical, morphological and molecular evidence that the immediate firing motoneurons are S-type motoneurons whereas the delayed firing ones are F-type motoneurons. First, immediate and delayed firing motoneurons are alpha-motoneurons and not gamma-motoneurons, since both of them receive proprioceptive inputs (VGlut1 contacts), express NeuN and have soma sizes in the alpha-motoneuron range (*Friese et al., 2009*). Second, the immediate firing motoneurons express Errβ but not Chodl as expected for S-type motoneurons (*Enjin et al., 2010*). Conversely, the delayed firing motoneurons do not express Errβ and those likely to innervate the largest fast-contracting motor units express Chodl as seen by *Enjin et al. (2010)*. Third, delayed firing motoneurons display, on average, larger input conductances, higher rheobases, and shorter AHPs than the immediate firing ones. This is in keeping with the electrical differences that have been observed between F-type and S-type motoneurons in adult cats (*Burke, 1981*; *Zengel et al., 1985*), rats (*Beaumont and Gardiner, 2002*; *Button et al., 2006*) and mice (*Manuel and Heckman, 2011*). Finally, the delayed firing motoneurons display a larger dendritic tree than the immediate firing motoneurons. Similar differences in the dendritic trees have been reported between F-type and S-type motoneurons in cats (*Burke et al., 1982*; *Cullheim et al., 1987*).

## Immediate vs delayed firing pattern: a factor not taken into account in previous studies of neonatal ALS mice

In most studies of neonate motoneurons, the distinction between the immediate and the delayed firing patterns has been overlooked (*Fulton and Walton, 1986*; *Vinay et al., 2000*; *Miles et al., 2002*). This was probably due to the fact that these studies used pulses of much shorter duration to elicit firing. Indeed, in the delayed firing motoneurons, the long latency of the first spike is only apparent on long-lasting stationary current pulses close to the rheobase (no spike would have been visible with this current intensity if a shorter pulse duration would have been used). If the pulses are short, higher current intensities are required to make the neuron fire, and the first action potential is fired with a shorter delay after the pulse onset. When the current was large enough, the discharge started shortly after the current onset (see *Figure 1—figure supplement 1*). In addition, since a slow current contribute to the delayed of firing (*Leroy et al., SfN abstract 2012*), long intervals are needed to allow it to recover their initial state before the next pulse. A rapid repetition of the test pulses might prevent the observation of the delayed firing. However, spinal motoneurons with a delayed discharge have been observed in *Pambo–Pambo et al. (2009)* who carefully checked the current intensity at which the motoneurons start to fire in response to a 0.5 s square pulse (see also *Russier et al. (2003)* for abducens motoneurons). Nonetheless, in these studies the latency of the first action potential was substantially shorter than in the present study, likely because the intensity to bring the cell to fire was slightly higher than the one that would have been required for a 5 s pulse duration. It is noteworthy that *Pambo–Pambo et al. (2009)* observed similar proportions of the two firing patterns.

In previous ALS studies, data coming from immediate and delayed firing motoneurons were pooled together when comparing motoneurons from mSOD1 mice and WT mice (*Bories et al., 2007*; *Amendola and Durand, 2008*; *Pambo–Pambo et al., 2009*; *Quinlan et al., 2011*). Since many morphological and electrophysiological properties differ between these two populations, the statistics for each property (average values, standard deviations) are heavily dependent on the proportion of immediate and delayed firing motoneurons present in each sample. If these proportions differ substantially in the WT and mSOD1 samples, this might obfuscate the conclusions. Therefore, one must be careful to separate these two motoneuron subtypes when comparing electrical and morphological properties between WT and mSOD1 neonatal mice.

## Origin of the intrinsic hyperexcitability in the immediate firing motoneurons in P6-P10 mSOD1 mice

The intrinsic hyperexcitability of mSOD1 immediate firing motoneurons is not due to a decrease in their input conductance. Indeed, the input conductance remains unchanged despite a reduction of the dendritic length of the mSOD1 immediate firing motoneurons. The relatively large overall size of the motoneuron dendritic tree in the second postnatal week probably limits the influence that morphological alterations have on the input conductance (*Elbasiouny et al., 2010*). Furthermore the reduction in dendritic length could be compensated by a decrease of the specific membrane resistivity. The average 5 mV hyperpolarization of the voltage threshold for spiking likely accounts for the reduction of the rheobase and thereby the intrinsic hyperexcitability of mSOD1 immediate firing motoneurons. *Quinlan et al. (2011)* observed an increase in the sodium persistent inward current in mSOD1 motoneurons during the second post-natal week. This increase in sodium persistent current can account for both the disappearance of MMOs and the hyperpolarization of the voltage threshold for spiking (*Iglesias et al., 2011*). In a modeling study, *Dai et al. (2002)* showed that spiking threshold hyperpolarization occurs when the sodium conductance increases or when the activation curve of the sodium channels is shifted toward hyperpolarized levels. They report that this effect on the spiking threshold occurs without changes in amplitude and width of action potentials, as in the present study. To a lesser extent, a reduction of the delayed rectifier conductance or a depolarization of its activation curve may also hyperpolarize the spiking voltage threshold (*Dai et al., 2002*).

In sharp contrast, we did not observe any change in intrinsic excitability in the motoneurons with the delayed firing pattern. This type of firing is due to the specific presence in these motoneurons of two potassium currents that act at two time scales: an A-current and a slowly activating and inactivating potassium current (*Leroy et al., SfN abstract 2012*). These two currents make the F-type motoneurons less excitable than the S-type motoneurons in neonates. Moreover, it is possible that these two currents prevent the spiking threshold to be hyperpolarized and thereby the rheobase to be decreased in mSOD1 neonates. However the delayed firing pattern is transient and disappears with age (*Russier et al., 2003*). In adults, the lower excitability of F-type motoneurons observed in WT animals is mainly due to the fact that they are much larger than the S-type motoneurons.

The origin of the hyperexcitability in the immediate firing motoneurons seems to be different from the mechanism at work in late embryonic motoneurons for which hyperexcitability arises from a decrease in input conductance (*van Zundert et al., 2008*; *Martin et al., 2013*). Numerical simulations suggest that the decrease in input conductance is due to a shortening of the terminal segments of embryonic spinal motoneurons which are much more electrotonically compact than in neonates (*Martin et al., 2013*). Interestingly, in spinal muscular atrophy, another motoneuron degenerative disease, spinal motoneurons also become hyperexcitable (*Mentis et al., 2011*). In this case, the increased excitability is caused both by an hyperpolarization of the spiking threshold and by a decrease in input conductance.

## Are the alterations of the dendritic tree caused by modifications of the synaptic activity?

The shortening of the dendritic tree of the immediate firing motoneurons might result from an increase in spontaneous synaptic activity occurring during the embryonic life (*Yvert et al., 2004*). Alterations in both the inhibitory and excitatory synaptic inputs impinging on mSOD1 motoneurons, as well as the properties of their postsynaptic receptors, have been reported in cultured motoneurons derived from embryos (*Carunchio et al., 2008*; *Chang and Martin, 2011*) as well as in motoneurons from neonates (*van Zundert et al., 2008*) and adults (*Jiang et al., 2009*; *Sunico et al., 2011*; *Wootz et al., 2013*). It has been shown that, during a critical developmental period, the morphology of the dendritic tree deeply relies on the synaptic activity received by the dendrites (*Spitzer, 2006*; *Cline and Haas, 2008*). More synaptic inputs can lead to a shortening of the dendrites (*Tripodi et al., 2008*). Such structural homeostatic response of the motoneurons might counterbalance the increase in synaptic activity to ensure that an appropriate level of input is achieved (*Tripodi et al., 2008*). The change in voltage threshold might also be a homeostatic regulation in response to synaptic hyperactivity. Since the frequency of both excitatory and synaptic spontaneous events are increased in early stages (*van Zundert et al., 2008*), it is possible that the net input to S-type motoneurons is shifted towards more inhibition. An excess of inhibition at early age might be

compensated by the hyperpolarization of the voltage threshold that increases the motoneuron excitability. As suggested by *van Zundert et al. (2008)* synaptic hyperactivity, dendritic shrinkage and intrinsic hyperexcitability might well be causally linked. The differential impact of the disease on the dendritic morphology of our two subpopulations of motoneurons suggests that synaptic alterations might be restricted to one subtype of motoneurons. Demonstrating this point will require, however, a preparation in which it is possible to preserve the integrity of spinal networks and identify the subtype of recorded motoneurons.

### Intrinsic hyperexcitability of mSOD1 motoneurons: detrimental or beneficial?

Our results show that only the S-type motoneurons display an intrinsic hyperexcitability in mSOD1 neonates. We further confirmed that the motoneurons displaying hyperexcitability were ALS resistant thanks to the expression pattern of MMP9 (*Kaplan et al., 2014*). On the other hand, F-type motoneurons vulnerable in ALS, are not hyperexcitable. We can therefore conclude that, contrary to the standard hypothesis (*Ilieva et al., 2009*), intrinsic hyperexcitabilty is not an early event that triggers degeneration of the motoneurons.

In mSOD1 mice, degeneration of the neuromuscular junctions of F-type motor units does not start before P50 whereas the S-type motor units do not degenerate (*Pun et al., 2006*; *Hegedus et al., 2008*). We may wonder whether the morphological and electrophysiological changes that we have observed perinatally in S-type motoneurons have a long term impact in adulthood and whether these early changes contribute to the survival of S-type motoneurons. The specific dendritic shrinkage of S-type motoneuron increases the size difference between S- and F- types motoneurons, and it has most likely a beneficial effect on S-type motoneurons. Indeed the smaller the total membrane surface, the smaller the metabolic demand to maintain this surface. Given the fact that motoneurons cope to an energetic issue in ALS that is caused by a disruption of the mitochondrial function (*von Lewinski and Keller, 2005*; *Ilieva et al., 2009*), a lesser energetic demand may increase their chance to survive. Specific hyperexcitability of S-type motoneurons might also contribute to motoneuron survival as recently suggested (*Saxena et al., 2013*). One might question whether an early hyperexcitability of S type motoneurons has a long-term benefit on their survival or whether the hyperexcitability has an impact only if it is still present in adults. We do not know whether S-type motoneurons remain hyperexcitable in adults. However, the whole population of adult spinal motoneurons are, on average, not hyperexcitable just prior to the onset of neuromuscular junctions degeneration (*Delestree et al., 2014*). Some of them even turn out to be hypoexcitable since they lose the capacity to discharge repetitively in response to stationary inputs. However, in *Delestree et al. (2014)*, S-type motoneurons (which represent only a small fraction of motoneurons) and F-type motoneurons were pooled together and the possibility that S-type motoneurons remain hyperexcitable in adults cannot be ruled out. Unlike neonates, adult S-type and F-type motoneurons cannot be distinguished on the basis of their discharge pattern. Only highly demanding in vivo experiments would allow distinguishing motor unit sub-types on the basis of their contractile properties. A selective intrinsic hyperexcitability of S-type motoneurons that persists in adults would strongly reinforce the assumption that hyperexcitability contributes by itself to the protection of these motoneurons.

## Materials and methods

### Ethical standards

The experiments were performed in accordance with European directives (86/609/CEE and 2010-63-UE) and the French legislation. They were approved by Paris Descartes University ethics committee (Permit Number: CEEA34.BLDI.065.12.). All surgery was performed under sodium pentobarbital anesthesia, and every effort was made to minimize suffering. 6 to 10 day-old high expressor line B6. Cg-Tg(SOD1-G93A)1Gur/J mice and their non-transgenic littermates of either sex were used (The Jackson Laboratory, RRID:IMSR_JAX:004435). Genotyping was performed following the protocol given by the Jackson Laboratory.

## Slice preparation

Mice were anesthetized using an intra-peritoneal 0.1 ml injection of pentobarbital 10% (5.5 mg/ml). Oblique slices were then prepared from the L3 to L5 spinal segments in order to keep a ventral root-let in continuity with the cord as described in *Lamotte d'Incamps et al. (2012)*. The slices were transferred into artificial cerebrospinal fluid (ACSF) containing (in mM): 130 NaCl, 2.5 KCl, 2 CaCl$_2$, 1 MgCl$_2$, 1 NaH$_2$PO$_4$, 26 NaHCO$_3$, 25 glucose, 0.4 ascorbic acid, 2 Na-pyruvate, bubbled with 95% O$_2$ and 5% CO$_2$ (pH 7.4).

## Electrophysiology

The recording chamber was continuously perfused with ACSF at a rate of 1–2 ml/min, at room temperature. The slices used were those containing a ventral rootlet of sufficient length to be mounted on a suction stimulation electrode: a glass pipette with a tip size adapted to the diameter of the rootlet (40–170 µm) and filled with ACSF. Patch pipettes had an initial open-tip resistance of 3–6 MΩ. The internal solution contained (in mM): 140 K-gluconate, 6 KCl, 10 HEPES, 1 EGTA, 0.1 CaCl$_2$, 4 Mg-ATP, 0.3 Na$_2$GTP. The pH was adjusted to 7.3 with KOH, and the osmolarity to 285–295 mOsm. An AxoClamp 700B (Molecular Device, Sunnydale, CA) amplifier was used for data acquisition. Whole-cell recordings were filtered at 3 kHz, digitized at 10 kHz using a CED 1401 and monitored using the Signal 5 software (Cambridge Electronic Design Limited). Bridge resistance was compensated in current-clamp mode. Liquid junction potential was not corrected in order to readily compare with previous studies.

## Neuron selection

We targeted large cells (long soma axis >20 µm) in the ventral horn under visual control using a video-camera (Scientifica, Uckfield, UK) and confirmed their motoneuron identity based on the recording of an antidromic action potential following stimulation of the ventral root. Single biphasic stimulation of the ventral rootlet (1–50 V, 0.1–0.3 ms) was used to elicit antidromic action potential in motoneurons. We retained for analysis motoneurons exhibiting a resting potential equal or below −50 mV and an overshooting action potential. Access resistance ranged from 8.5 to 20 MΩ. Motoneurons were discarded from analysis if series resistance or resting potential varied more than 5 MΩ or 10 mV throughout the recording period. These criteria are similar to previous studies (*Pambo–Pambo et al., 2009*; *Quinlan et al., 2011*). Three delayed firing motoneurons displayed unusualy large conductances (123, 151 and 153 nS). When excluding them our measurements fell in the same range and had the same variability than in previous studies (*Pambo–Pambo et al., 2009*; *Quinlan et al., 2011*). However, we chose to maintain these three delayed firing motoneurons in our calculations and analysis.

## Intracellular labeling and 3D-reconstruction

Some of the electrophysiologically characterized motoneurons were filled with neurobiotin in order to study the anatomy of their dendritic tree. The intracellular solution was supplemented with 2% neurobiotin (Vector Labs, Burlingame, CA) and the motoneurons were recorded for at least 30 min to allow diffusion of the dye. After carefully removing the electrode from the cell, the slice was bathed in phosphate-buffered saline (PBS) with 4% paraformaldehyde for 1 hr. Blocking solution containing 0.1% of bovine serum albumin and 0.1% Triton X-100 (Sigma–Aldrich, St. Louis, MO) in PBS was applied for 1 hr. Slices were incubated overnight at 4°C with streptavidin-Cy3 conjugated antibody (Sigma–Aldrich) diluted at 1/500 in the blocking solution, washed three times in PBS and then mounted with Fluoromount (Sigma–Aldrich). Acquisition was performed on a confocal microscope LSM 710 (Carl Zeiss, Oberkochen, Germany) and the dendritic tree of the motoneuron was reconstructed using Neurolucida software (MBF Bioscience Williston, VT, RRID:nif-0000-10294). Because of the slicing procedure, parts of the dendritic trees were missing and motoneurons reconstructions are therefore partial. Analysis of the dendritic trees included only the radial dendrites that remained in the same plane as the slice and did not plunge deeper than 50 µm below the surface of the slice. However, the number of reconstructed primary dendrites (and thereby dendritic trees) per motoneuron was similar between immediate and delayed firing motoneurons and we could therefore readily compare those radial dendrites in the two subtypes.

## Immunostaining

A sample of motoneurons was electrophysiologically characterized, filled with neurobiotin and subsequently immunostained for NeuN, VGglut1, Errβ or MMP9. Following fixation, the slices were incubated in blocking solution (see above). Then, the slices were incubated overnight at 4°C in the blocking solution supplemented with 1:500 rabbit anti-NeuN (Cat# ABN78; EMD Millipore, Billerica, MA, RRID:AB_10807945), 1:4000 guinea-pig anti-VGlut1 (Cat# AB5905; EMD Millipore, RRID:AB_2301751) or 1:500 mouse anti-Errβ (Cat# PP-H6705-00; R&D Systems, Minneapolis, MN, RRID:AB_2100412) After three washes in PBS, 1:500 streptavidin-Cy3 and 1:500 of the appropriate secondary antibody were applied in blocking solution during 3 hr at room temperature. We used the following secondary antibodies: anti-rabbit Alexa 488-conjugated (Cat# 111-545-003; Jackson Immunoresearch, West Grove, PA), anti-guinea-pig Alexa 647-conjugated (Cat# 106-605-003; Jackson Immunoresearch) and anti-mouse CF633-conjugated (Cat# SAB4600333; Sigma–Aldrich). For MMP-9 labelling, neurobiotin was used at 0.2% and the blocking solution contained 3% BSA, 0.5% Triton and 5% Horse Serum. Slices were subsequently incubated with 1:500 goat anti-MMP9 (Cat# M9570; Sigma–Aldrich, RRID:AB_1079397) and then with 1:1000 anti-goat Cy3-conjugated (Cat# 705-165-003; Jackson ImmunoResearch) and 1:500 streptavidin Cy2-conjugated (Cat# 016-220-084; Jackson ImmunoReserach). All slices were mounted and imaged as described above.

## In situ hybridization

In another sample of motoneurons electrophysiologically characterized and filled with neurobiotin (see above), we performed chondrolectin in situ hybridization as described in *Enjin et al. (2010)*. Chondrolectin probes (Genebank number NM_139134.3) were produced from commercial cDNA (Source BioScience, Nottingham, UK), using T3 RNA polymerase in the presence of digoxigenin-11-UTP (Roche Diagnostics, Basel, Switzerland). Slices were washed with PBT (PBS supplemented with 0.1% Tween-20, Sigma–Aldrich) followed by treatment with 0.5% Triton X-100. Slices were then post-fixed in 4% formaldehyde followed by prehybridization in hybridization buffer (50% formamide, 5× saline-sodium citrate [SSC], pH 4.5, 1% sodium dodecyl sulphate [SDS], 10 mg/ml tRNA [Life Technologies, Carslbad, CA], 10 mg/ml heparin [Sigma–Aldrich] in PBT). The probe (300 ng/ml) was heat-denatured before starting the overnight hybridization (20–22 hr) at 63°C. Overnight hybridization was followed by sequential washes with wash buffer 1 (50% formamide, 5× SSC, pH 4.5 and 1% SDS in PBT) followed by buffer 2 (50% formamide, 2× SSC, pH 4.5, and 0.1% Tween-20 in PBT) at 63°C to remove unbound probe. Slices were then washed in 0.1% Tween-20 Tris-buffered saline followed by incubation in 1% blocking reagent (Roche Diagnostics). Then the slices were incubated overnight at 4°C with 1:5000 diluted anti-digoxigenin alkaline phosphatase-conjugated antibody (Roche Diagnostics). Hybridized probes were vizualised using SIGMAFAST Fast Red TR/Naphthol AS-MX (Sigma–Aldrich). After hybridization, neurobiotin was revealed by washing the slices in PBS followed by PBS-T-G (PBS, 0.25% Triton X-100, 0.25% Gelatin). Slices were then incubated with 7.5 µg/ml Cy2-conjugated streptavidin (Jackson Immunoresearch) diluted in PBS-T-G for 2 hr at room temperature.

## Data analysis

### Electrophysiology

Analysis of the recordings was performed using custom programs in Signal 5 (Cambridge Electronic Design Limited, Cambridge, UK). Input conductance was the inverse of the slope of the I-V curve obtained by injecting small 500 ms pulses of currents (−100 pA to +20 pA, 30 pA steps repeated 10 times). The rheobase was searched by applying a series of 5 s square pulses of increasing intensity. The time between the end of a pulse to the beginning of the following one was 20 s. This long interval allowed the slow currents to recover their initial state before the next pulse. Given this long interval interpulse we had to limit the number of pulses tested. The pulse intensity was increased, by 50 pA steps, from 0 to the intensity that elicits the firing of at least one action potential. However when the rheobase was higher than 0.9 nA, the steps were increased to 100 pA. This step searching protocol made it hard to systematicaly measure the exact value of current required to elicit only a single action potential and we would often transition from a sub-threshold response to a train of action potentials between two consecutives steps. This minimal current that elicited the cell firing (a single potential or a train) was therefore considered as the rheobase. We also measured the recruitment

current for which the first action potential was fired during a 0.1 nA/s current ramp (average of three trials). Note that the recruitment currents are very similar to the rheobases assessed with the pulses (*Table 1*). We measured the voltage spiking threshold on the first spike of the 5 s pulses as the voltage for which the first derivative value went over 10 mV/ms. Single action potentials were elicited by 1 ms square pulses (1–10 nA). Their height, width at half-amplitude as well as the relaxation time constant of their AHP were measured on the average of 30 successive trials. The relaxation time constant of the AHP was determined using a mono-exponential fit.

## Anatomy

A brightfield image of the soma of each recorded motoneuron was systematically taken and the soma area was measured using FIJI (*Schindelin et al., 2012*). Morphological analysis of the dendritic tree of labeled motoneurons was conducted with the Neurolucida software and summary data was collected in Microsoft Excel (Microsoft, Redmond, WA). We measured the following parameters: primary dendrites number (including only the reconstructed dendrites, see above), branching points number (number of dendritic bifurcations), total dendritic length (sum of the lengths of all reconstructed dendrites), dendritic paths (trajectory from the tip of every terminal segments to the soma), and terminal segments length.

## Statistics

We used two-tailed *Mann–Whitney U* tests to assess the difference between two properties while two-tailed *Fischer* exact tests were used on contingency tables. The *Kolmogorov–Smirnov* bilateral test was used to compare distributions in *Figure 8*. The slopes of the linear regressions of *Figures 6* and *8* were compared with the use of a *t* test, after verification that the residues of the regressions were normally distributed (*Shapiro–Wilk* tests). Further verification using *Chow* test (*Chow, 1960*; *Studinger et al., 2007*) led to the same results. The regressions of *Figure 6* were constrained to a null intercept (to satisfy the law I = G.U where I is the rheobase, G the input conductance and U, the slope of the regression has the dimension of a voltage). U is close to the mean of $\Delta V = (V_{threshold\ for\ spiking} - V_{rest})$ for the immediate firing motoneurons but not for the delayed firing ones. Indeed, for the delayed firing motoneurons, the membrane potential is depolarized during the long pulses. As a result $V_{threshold}$ for the first spike is more depolarized than it would have been if the firing had occurred at the pulse onset. Accordingly, U is smaller than the mean of $\Delta V$ for the delayed firing motoneurons. All tests and analysis were undertaken with R (*Team RC, 2013*) version 3.0.2 and the real statistics Excel resource pack (*Zaiontz, 2014*). Data in the text and tables are expressed as mean ± SD with the range and the number of observations. Source data can be found in the *Supplementary file 1*.

# Acknowledgements

The authors wish to thank Pr Philippe Ascher and Dr Marin Manuel for helpful comments on the work and careful scrutinizing of the manuscript. We thank Pr CJ Heckman and Dr Kathy Quinlan who commented on an early version of the manuscript. We also thank Pr Marc Abitbol and Pr Klas Kullander for their advice on in situ hybridization, Pr Klas Kullander for providing the chondrolectin probe and Pr Martyn Goulding for his advice regarding Errβ immuno-histochemistry. Financial supports provided by the Agence Nationale pour la Recherche (HYPER-MND, ANR-2010-BLAN-1429-01), the NIH-NINDS (R01NS077863), the Thierry Latran Fundation (OHEX Project) and Target ALS are gratefully acknowledged. Felix Leroy was recipient of a "Contrat Doctoral" from the Ecole Normale Supérieure, Cachan.

# Additional information

## Funding

| Funder | Grant reference number | Author |
|---|---|---|
| Agence Nationale de la Recherche | ANR-2010-BLAN-1429-01 Hyper-MND | Daniel Zytnicki |

| National Institute of Neurological Disorders and Stroke (NINDS) | R01NS077863 | Daniel Zytnicki |
| Thierry Latran Foundation under the aegis of Fondation de France | Thierry Latran Foundation - OHEX project | Daniel Zytnicki |
| Target ALS | Excitability Consortium Project | Daniel Zytnicki |
| Ecole Normale Supérieure de Cachan | Graduate Student Fellowship | Félix Leroy |

The funders had no role in study design, data collection and interpretation, or the decision to submit the work for publication.

## Author contributions

FL, Conception and design, Acquisition of data, Analysis and interpretation of data, Drafting or revising the article; BLI, Acquisition of data for the MMP9 labelling, Conception and design, Analysis and interpretation of data, Drafting or revising the article; RDI-M, RDI-M, Acquisition of data for the in situ hybridization and MMP9 labelling; DZ, Conception and design, Analysis and interpretation of data, Drafting or revising the article

## Author ORCIDs

Boris Lamotte d'Incamps, http://orcid.org/0000-0003-4221-7526

## Ethics

Animal experimentation: The experiments were performed in accordance with European directives (86/609/CEE and 2010-63-UE) and the French legislation. They were approved by Paris Descartes University ethics committee (Permit Number: CEEA34.BLDI.065.12.). All surgery was performed under sodium pentobarbital anesthesia, and every effort was made to minimize suffering.

## Additional files

### Supplementary files

• Supplementary file 1. Source data for *Tables 1* and *2* and *Figures 6* and *8*. Each sheet begins with a description of the properties measured and where they are displayed in the article.

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
