## [Decision Letter]

Thank you for sending your work entitled “Early intrinsic hyperexcitability does not contribute to motoneuron degeneration in amyotrophic lateral sclerosis.” for consideration at *eLife*. Your article has been favorably evaluated by Eve Marder (Senior editor), and 3 reviewers, one of whom, Ole Kiehn, is a member of our Board of Reviewing Editors, and another of whom, Ken Rose, has agreed to reveal his identity.

The Reviewing editor and the other reviewers discussed their comments before we reached this decision, and the Reviewing editor has assembled the following comments to help you prepare a revised submission.

The sequence of cellular and molecular events leading to the dysfunction and death of motor neurons in ALS is not understood. A prevailing hypothesis that has been suggested from experiments in numerous previous studies is that early motor neuron hyperexcitability invariably causes late motor degeneration in ALS. Despite the attractive logic of this hypothesis, all of these studies suffer from a potentially fundamental shortcoming; that recordings have not distinguished between slow and fast motor neurons. Slow motor neurons innervating slow contracting muscle fibers are much more resistant to degeneration than motor neurons innervating fast contracting muscle fibers during development of ALS. Thus, the ability to distinguish slow and fast motor neurons during intracellular recordings has represented an acute and pressing problem that has been solved in the present study. In the present study the authors take advantage of electrophysiological, molecular and anatomical techniques to identify slow and fast motor neurons in young wild type mice and mSOD1 mice. The data provide compelling evidence that slow and fast motor neurons can be distinguished in both groups of mice. The results demonstrate a clear and marked change in the excitability of motor neurons in mSOD1 mice that depends on motor neuron type. However, surprisingly the excitability of motor neurons innervating fast contracting muscle fibers does not change. In contrast the excitability of motor neurons innervating slow contracting muscle fibers increases. Unlike previous studies of motor neurons in ALS models the study links the motor excitability to the different subtypes of motor neurons. The study, therefore, provides a significant advance. Despite the study's significant merits there are a number of issues that need to be considered as outlined below: 1) From the Introduction section: The observation of “delayed firing” in approximately two thirds of wild type motor neurons is surprising and puzzling. Previous reports have only described the “immediate” type of firing (Fulton and Walton, 1986; Vinay et al, 2000; Pflieger et al, 2002; Miles et al, 2005) or delayed firing that is less pronounced (Pampo-Pambo et al.). Are these differences caused by a sampling bias, do they have to do with the place of sampling, or do they reflect the real relationship between the numbers of fast and slow motor neurons in the spinal cord at this age? A much better and in depth discussion of this finding with relationship to previous literature is needed; in particular a discussion of how the firing feature is causally related to hyperexcitability which may imply that the hyperexcitability of these cells also is transient and may disappear with age.

2) Some motor neurons fired at extremely low Vthr (i.e. below -60mV, Fig.6A2). How is that explained? The authors do not report any data on the resting membrane potential for the two populations of motor neurons from either wild-type or ALS motor neurons. Was there any difference between wild type and ALS motor neurons? This information is needed to assess the two populations. Furthermore, the criteria for selection of motor neurons are set to a resting potential at -65mV or lower. The study by Pambo-Pambo and colleagues (2009) reported that ALS motor neurons exhibited elevated resting potential (-55mV) compared to wild type motor neurons. Is it possible that the authors in the present study might have excluded a significant proportion of ALS motor neurons from their study by setting too stringent criteria? A clarification of these issues with relationship to the previous literature is absolutely essential to ensure that there has not been a sampling bias that might jeopardize the general conclusions.

3) There are a number of terms that need to be defined more clearly and explained in the text. For example, it is unclear how the authors define rheobase. Is it the minimum current to induce a single action potential, or is it the current to induce repetitive firing? The authors do not provide any information, either in terms of recordings or in the manuscript how precisely this analysis was performed. This clarification has a significant weight in the interpretation of the results and the authors should state the method they have used. The discussion about primary range and sub-primary range is very confusing for a person who is not familiar with the terminology and it should be clarified in the text. Moreover the relevance of the mixed mode oscillations in the context of motor neuron pathology is unclear and a better discussion of this phenomenon is needed.

4) There is a high degree in variability in certain physiological parameters. The input conductance (from Table 1, for example, the “delayed” firing ranges from 10 to 151nS), the rheobase (again from Table 1., in the “delayed” firing ranges from 0.3 to 2.8pA) and the Vthr (similarly, it ranges from -62 to -31mV). This variability in parameters is large compared with other previous work and the authors should provide an explanation. Is it possible that the slicing procedure may be responsible for this variability since a large extent of the motor neuron dendritic tree is compromised? Of course it is also possible that previous studies under-reported their variability, but this issue needs to be confronted.

5) There is a mismatch between the rheobase values and recruitment threshold values presented in Table 1 (i.e. in pA) and those presented in the Figures and graphs, in which the authors present the numbers are of magnitude higher (i.e. in nA). The authors should correct this discrepancy (i.e. pA or nA).

6) In the Results section the authors state: 'However, in mSOD1 mice, the relationship between the rheobase and the input conductance is affected selectively in immediate firing motoneurons (Figure 6) whereas it is virtually unchanged in delayed firing motoneurons (Figure 6). However this difference in the rheobases of slow motor neurons is only apparent for motor neurons with input conductances greater than 40 nS. Is it possible that ALS may increase the excitability of a subset of slow motor neurons; those with larger conductances? This might be apparent by performing a regression analysis of the data to determine directly the 'relationship' between rheobase and input conductance and thereby expand the present analysis.

7) Figure 7 is essential for appreciating the changes in excitability. It would benefit from being more clearly presented with better examples of data from the SOD and wild-type mice. First of all the different motor neuron types and mice types should be labeled in the figure. Secondly, MNs should be chosen so that the records clearly show that the amplitude for activation is the same or lower for MNs in SOD mice (this is not the case for the cells shwon in C1 and D1). Finally, all the X-axes and Y-axes should be the same so it is easy to compare the firing frequency and the amplitude of current injection.

8) The change in the dendritic morphology of the immediate firing motor neurons in SOD mice is not explained and might be a cause of synaptic pruning. Previous experiments from Caroni's lab have suggested that there is a reduction in inhibitory synapses onto ALS-affected motor neurons. Some comments on these findings would be appropriate. Also, a discussion of whether such a pruning of synaptic input will change the threshold for axon potential generation will be appropriate.

9) The Discussion section should be expanded to include a discussion of developmental issues as mentioned above, including the possibility that the changes observed perinatally may be part of processes that maintain homeostasis. A better discussion of what functional role the hyperexcitability might have including the issue of the “protective” property of hyperexcitability in the context of ALS is also needed.

[Editors’ note: further revisions were requested prior to acceptance, as shown below.]

Thank you for sending your revised work entitled “Early intrinsic hyperexcitability does not contribute to motoneuron degeneration in amyotrophic lateral sclerosis.” for consideration at *eLife*. Your article has been favorably evaluated by Eve Marder (Senior editor) and 3 reviewers, one of whom is a member of our Board of Reviewing Editors. With one significant exception, you have systematically addressed all of the concerns/suggestions for revisions, either by including new data analysis, or extensively revising the text and figures, or by offering well-reasoned rebuttals. Some changes to the manuscript are still needed before it can be published. These are outlined below.

Despite the improved match between the values previously for input conductances and the results of the present study by excluding data from 3 cells with unusually large conductances, there is no support for this decision to exclude these data. The decision is arbitrary and, given that all cells satisfied the same inclusion criteria, not justified. These data should therefore be included with a brief description noting that impact of excluding the so-called outliers. If the rheobase of these cells is known, these data should also be added to the graph in Figure 6 or the exclusion of these cells should be noted.

---

## [Author Response]

*1) From the Introduction section: The observation of “delayed firing” in approximately two thirds of wild type motor neurons is surprising and puzzling. Previous reports have only described the “immediate” type of firing (Fulton and Walton, 1986; Vinay et al, 2000; Pflieger et al, 2002; Miles et al, 2005) or delayed firing that is less pronounced (Pampo-Pambo et al.). Are these differences caused by a sampling bias, do they have to do with the place of sampling, or do they reflect the real relationship between the numbers of fast and slow motor neurons in the spinal cord at this age? A much better and in depth discussion of this finding with relationship to previous literature is needed*.

The differences between our study and previous ones mainly arise from the fact that we looked at the discharge that was elicited when applying long duration pulses at a liminal intensity, in contrast with previous studies that used shorter pulses. We made a new Figure (Figure 1—figure supplement 1) to make the point clearer. In the delayed firing motoneurons, upon injection of rheobase current, the first spike appeared at the end of the pulse with a long delay (no spike would have been visible with this current intensity if a shorter pulse duration would have been used). However the delay decreased when the current intensity increased. When the current was large enough the discharge started shortly after the current onset. These explanations have now been added in the first paragraph of the Results section and in the Material and methods section explaining with greater details how we measured the rheobase. The Discussion section was expanded to explain that the delayed firing pattern might have been overlooked if the current intensity is too high. In addition, since slow inactivating currents contribute to the delayed firing (Leroy et al. SfN abstract 2012), long intervals are needed between successive test pulses to allow these slow currents to recover their initial state before the next pulse. A rapid repetition of the test pulses might prevent the apparition of the delayed firing. Finally, as now discussed, we suggest that [35] have shorter delays than in the present work because they were using shorter pulse duration that required slightly higher current to elicit firing in that time window. It is noteworthy that [35] observed a similar proportion of motoneurons exhibiting the delayed firing and the immediate firing patterns as in the present study.

*… in particular a discussion of how the firing feature is causally related to hyperexcitability which may imply that the hyperexcitability of these cells also is transient and may disappear with age*.

The delayed firing pattern is caused by the presence of two potassium currents that act at two time scales: an A-current and a slowly inactivating current. They contribute to make the delayed firing motoneurons less excitable than the immediate firing ones in neonates (Leroy et al. SfN abstract 2012). However the delayed firing pattern is transient and disappears in adults (39; 29). In adults, the lower excitability of the F-type motoneurons is mainly due to the fact that they are much larger than the S-type motoneurons. This has been added in the Discussion section.

*2) Some motor neurons fired at extremely low Vthr (i.e. below -60mV, Fig.6A2). How is that explained*?

This is because we used an off-line correction for the liquid junction potential as stated in the Material and methods section in contrast to the previous work where motoneurons were recorded with similar solutions that should have called for a similar off-line correction. We computed the theoretical liquid junction potentials in [35] and in [38] using the JPCalcW module (Barry et al., 1994) in pClamp and found that they were close to our liquid junction potential. In order to facilitate the comparison with these works we have decided in this revised version to not use any longer this off-line correction.

*The authors do not report any data on the resting membrane potential for the two populations of motor neurons from either wild-type or ALS motor neurons. Was there any difference between wild type and ALS motor neurons? This information is needed to assess the two populations. Furthermore, the criteria for selection of motor neurons are set to a resting potential at -65mV or lower. The study by Pambo-Pambo and colleagues (2009) reported that ALS motor neurons exhibited elevated resting potential (-55mV) compared to wild type motor neurons. Is it possible that the authors in the present study might have excluded a significant proportion of ALS motor neurons from their study by setting too stringent criteria? A clarification of these issues with relationship to the previous literature is absolutely essential to ensure that there has not been a sampling bias that might jeopardize the general conclusions*.

The resting membrane potential is now added in the Table 1. We have added (Results section) that there is no difference between the two populations in WT animal. There was also no difference between WT and ALS motoneurons as now stated in the Results section. Note that Pambo-Pambo (2009) reported an elevation of the resting potential in motoneurons of G93A-Low expressor mice but not on G85R mice. In the present work we have used the more classical G93A-High expressor mice.

We do not believe that we have excluded a significant proportion of ALS motoneurons by setting too stringent selection criteria. Actually the criteria for the selection of motoneurons are set to a resting potential at -50 mV before the off-line correction. Once again no correction of the membrane potential was made in either [35] or in [38] although they used similar solutions to perform their recordings. Our selection criterion were the following: we kept every cell that was identified as a motoneuron on the basis of the observation of an antidromic action potential, that displayed a resting potential equal or more hyperpolarized than -50 mV (non corrected potential) and that displayed an overshooting action potential and an access resistance from 8.5 to 20.0 MΩ. These criteria were similar to [38] (except for the antidromic identification that guaranteed us that we were recording a motoneuron). A new section entitled “Neuron selection” has been added in the Methods to clarify these issues. The average resting membrane potentials are now added in Table 1. Our data are close to [38] and [35] (respectively -60 mV and -58 mV).

*3) There are a number of terms that need to be defined more clearly and explained in the text. For example, it is unclear how the authors define rheobase. Is it the minimum current to induce a single action potential, or is it the current to induce repetitive firing? The authors do not provide any information, either in terms of recordings or in the manuscript how precisely this analysis was performed. This clarification has a significant weight in the interpretation of the results and the authors should state the method they have used*.

We better defined rheobase in the Materials and methods section (Electrophysiology sub-section). That was the minimum current that elicited at least a single action potential. Now we also explained in the same section that it was difficult to find the current that elicited just one spike because the searching procedure for the rheobase used pulses that were incremented by 50 pA steps (as in Pambo-Pambo 2009) when the rheobase was lower than 900 pA and then the steps were increased to 100 pA when rheobases were higher. This was achieved because we had to wait 20 seconds between two successive pulses in order to allow the slow currents to recover their initial state (if we have been using shorter intervals between the pulses we would have missed the delayed firing pattern) and this long interpulse duration prevents us to make too many sweeps. However, we have been sometime able to find the current that elicited a single action potential as in Figure 1—figure supplement 1. In other cases the motoneuron fired a train. In addition we also assessed the rheobase on slow ramps of currents (recruitment current). In this case the current intensity was continuously increasing and we measured the current (recruitment current) at which the motoneuron fired the first spike. The recruitment current is another way to measure the rheobase, and in fact lead to the same results as with the current pulses (see Table 1). The decrease in rheobase current for the immediate firing motoneurons in mSOD1 mice has thereby been demonstrated with two different methods. The Results section has been rewritten to better stress this point.

*The discussion about primary range and sub-primary range is very confusing for a person who is not familiar with the terminology and it should be clarified in the text. Moreover the relevance of the mixed mode oscillations in the context of motor neuron pathology is unclear and a better discussion of this phenomenon is needed*.

We clarified this issue. First, a definition of MMOs and sub-primary range is now given in the Results. We simplified a lot and reorganized the Figure 7 in order to better show the MMOs. We then explained within more details that MMOs reflect the excitability state of the cell. Indeed it has already been demonstrated that MMOs are caused by a low excitability state due to a slow inactivation of the sodium current that induces a relative deficit of sodium current with respect to potassium (20). Here we demonstrated that the immediate firing motoneurons are largely lacking MMOs in mSOD1 mice whereas there are present in WT mice. This further demonstrates that the excitability of immediate firing motoneurons has increased in WT mice. In sharp contrast delayed firing motoneurons display MMOs both in mSOD1 and WT mice indicating again an unchanged excitability. The last paragraph of the Results was fully rewritten to explain this issue.

*4) There is a high degree in variability in certain physiological parameters. The input conductance (from*Table 1*, for example, the “delayed” firing ranges from 10 to 151nS), the rheobase (again from*Table 1*, in the “delayed” firing ranges from 0.3 to 2.8pA) and the Vthr (similarly, it ranges from -62 to -31mV). This variability in parameters is large compared with other previous work and the authors should provide an explanation. Is it possible that the slicing procedure may be responsible for this variability since a large extent of the motor neuron dendritic tree is compromised? Of course it is also possible that previous studies under-reported their variability, but this issue needs to be confronted*.

We added an extra criterion compared to previous studies: the antidromic activation upon stimulation of the motor axon in the ventral rootlet, which guarantees that we record motoneurons. However, we had 3 outliers: 2 delayed firing motoneurons in WT mice (input conductances 124 and 151 nS) and 1 delayed firing motoneurons in SOD1 mice (input conductance 153 nS) that we have now removed from our statistics in Table 1. The ranges and the standard deviations of the tested parameters are then close to the ones of other studies. For instance, in WT motoneurons, the standard deviations (that we computed by multiplying the standard errors reported in previous works by the square root of the number of observations) were 20 nS for the input conductance and 0.57 nA for the rheobase in [35] and 7 mV for spiking voltage threshold in [38].

*5) There is a mismatch between the rheobase values and recruitment threshold values presented in*Table 1*(i.e. in pA) and those presented in the Figures and graphs, in which the authors present the numbers are of magnitude higher (i.e. in nA). The authors should correct this discrepancy (i.e. pA or nA)*.

The unit has been corrected in Table 1 (nA everywhere).

*6) In the Results section the authors state: 'However, in mSOD1 mice, the relationship between the rheobase and the input conductance is affected selectively in immediate firing motoneurons (*Figure 6*) whereas it is virtually unchanged in delayed firing motoneurons (*Figure 6*). However this difference in the rheobases of slow motor neurons is only apparent for motor neurons with input conductances greater than 40 nS. Is it possible that ALS may increase the excitability of a subset of slow motor neurons; those with larger conductances? This might be apparent by performing a regression analysis of the data to determine directly the 'relationship' between rheobase and input conductance and thereby expand the present analysis*.

We have expanded the analysis by performing a regression analysis of the data to determine the relationship between rheobase and input conductance. The slopes of the linear regressions have now been added in Figure 6. The regressions were constrained to a null intercept (to satisfy a law of the form I = GU). The slopes of the regressions were compared with the use of a t-test, after verification that the residues of the regressions are normally distributed (Shapiro-Wilk tests). This test supports the conclusion that the relationships differ between immediate firing motoneurons from WT and mSOD1 mice (Figure 6) whereas they are similar between delayed firing motoneurons from WT and mSOD1 mice (Figure 6). Further verifications were made using the Chow test (10). First, this structural test showed that the behavior of slow motoneurons with input conductances smaller than 40nS was not significantly different than the behavior of slow motoneurons with input conductances greater than 40nS (p=0.1). Second, the Chow test also allowed us to verify the conclusions of the t-test that we made on the slopes of the linear regressions showed on Figure 6. The Chow test led to the same results as the t-test. The tests results have been added in the Results section and the tests are explained in the Methods section (Data analysis, Statistics).

*7)*Figure 7*is essential for appreciating the changes in excitability. It would benefit from being more clearly presented with better examples of data from the SOD and wild-type mice. First of all the different motor neuron types and mice types should be labeled in the figure. Secondly, MNs should be chosen so that the records clearly show that the amplitude for activation is the same or lower for MNs in SOD mice (this is not the case for the cells shwon in C1 and D1). Finally, all the X-axes and Y-axes should be the same so it is easy to compare the firing frequency and the amplitude of current injection*.

Figure 7 has been substantially simplified in order to help the readers. We now show only an expansion of the beginning of the discharge in response to the slow current ramp. This allows to better show the oscillations (indicated by arrowheads) that occur between spikes. Small oscillations in between full spikes characterize MMOs. On Figure 7 we now fully labelled delayed and immediate/mSOD1 and WT as suggested by the reviewers.

In addition, we have made a new figure (Figure 7—figure supplement 1) that shows the full responses during the slow current ramps and the corresponding F-I relationships. We also clearly labelled each panel (delayed and immediate / mSOD1 and WT). It was confusing to present the immediate firing motoneurons in the first row and without labelling, so we apologize about that. Indeed the recruitment current is lower in immediate vs. delayed firing motoneurons, and it is lower in immediate firing mSOD1 motoneuron compared to immediate firing WT motoneurons. We now plot the F-I relationships using the same axes for each firing patterns.

8) The change in the dendritic morphology of the immediate firing motor neurons in SOD mice is not explained and might be a cause of synaptic pruning.

We suggest that the change of dendritic morphology is caused by an increase in spontaneous synaptic activity during embryonic life (52). This causes a structural homeostasis to counterbalance the increase of synaptic activity ensuring that an appropriate level of input is achieved (43; 11; 47). The Discussion section “Are the alterations of the dendritic caused by modifications of the synaptic activity” has been reorganized and expanded to highlight this issue.

*Previous experiments from Caroni's lab have suggested that there is a reduction in inhibitory synapses onto ALS-affected motor neurons. Some comments on these findings would be appropriate. Also, a discussion of whether such a pruning of synaptic input will change the threshold for axon potential generation will be appropriate*.

We do not think that a synaptic pruning will change the voltage threshold in neonates since it occurs much later during the animal life. To our knowledge, the reduction in inhibitory synapses occurs in adulthood at early-symptomatic age when dendritic vacuolization occurs (45). This is a late process that probably contributes to the excitotoxicity of spinal motoneurons.

However, [38] observed an increase of the sodium persistent current in motoneurons of mSOD1 mice. Such an increase of sodium persistent current can account both for the disappearance of the MMOs and for the hyperpolarization of the spiking voltage threshold as we previously shown (Iglesias et al). This is now stated in the Discussion section “Origin of the intrinsic excitability…” Moreover, we explain later on during the Discussion that this change in voltage threshold might be a homeostatic regulation in response to synaptic hyperactivity. Since the frequency of both excitatory and inhibitory synaptic spontaneous events are increased in early stages (Van Zundert et al. 2008), it is possible that the net input to S-type motoneurons is shifted towards more inhibition. An excess of inhibition at early age might be compensated by the hyperpolarization of the voltage threshold that increases the motoneuron excitability. In our study, we witness dendritic shrinking and intrinsic hyperexcitability, this opens the possibility that they result from an increase in synaptic activity. As suggested by Van Zundert et al. (2008) synaptic hyperactivity, dendritic shrinking and intrinsic hyperexcitability might well be causally linked.

*9) The Discussion section should be expanded to include a discussion of developmental issues as mentioned above, including the possibility that the changes observed perinatally may be part of processes that maintain homeostasis. A better discussion of what functional role the hyperexcitability might have including the issue of the “protective” property of hyperexcitability in the context of ALS is also needed*.

We now discuss the idea that, during the development, structural changes and hyperpolarization of the spiking voltage threshold are regulatory mechanisms that maintain structural and excitability homeostasis of motoneurons when spinal networks are hyperactive. The last section of the Discussion has been expanded to discuss how both the morphological changes and the intrinsic hyperexcitability might contribute to selectively protect S-type motoneurons in ALS.

[Editors’ note: further revisions were requested prior to acceptance, as shown below.]

*Despite the improved match between the values previously for input conductances and the results of the present study by excluding data from 3 cells with unusually large conductances, there is no support for this decision to exclude these data. The decision is arbitrary and, given that all cells satisfied the same inclusion criteria, not justified. These data should therefore be included with a brief description noting that impact of excluding the so-called outliers. If the rheobase of these cells is known, these data should also be added to the graph in*Figure 6*or the exclusion of these cells should be noted*.

We restored the 3 motoneurons in Table 1 as well as in the global source data file. The rheobase of two of these motoneurons is unknown; they are therefore not plotted in Figure 6. The third motoneuron is plotted in the Figure 6. We had already taken this motoneuron into account to perform the regression. In the Method section, we briefly comment on the impact of excluding these 3 delayed firing motoneurons with unusually large conductances.